# Why Some Models Resist Unlearning: A Linear Stability Perspective

## Abstract

Machine unlearning—the ability to erase the effect of specific training samples without retraining from scratch—is critical for privacy, regulation, and efficiency. However, most progress in unlearning has been empirical, with little theoretical understanding of when and why unlearning works. We tackle this gap by framing unlearning through the lens of asymptotic linear stability to capture the interaction between optimization dynamics and data geometry. The key quantity in our analysis is data coherence - the cross-sample alignment of loss-surface directions near the optimum. We decompose coherence along three axes: within the retain set, within the forget set, and between them, and prove tight stability thresholds that separate convergence from divergence. To further link data properties to forgettability, we study a two-layer ReLU CNN under a signal-plus-noise model and show that stronger memorization makes forgetting easier: when the signal-to-noise ratio (SNR) is lower, cross-sample alignment is weaker, reducing coherence and making unlearning easier; conversely, high-SNR, highly aligned models resist unlearning. For empirical verification, we show that Hessian tests and CNN heatmaps align closely with the predicted boundary, mapping the stability frontier of gradient-based unlearning as a function of batching, mixing, and data/model alignment. Our analysis is grounded in random matrix theory tools and provides the first principled account of the trade-offs between memorization, coherence, and unlearning.

## 1 Introduction

Machine unlearning – the ability to erase specific training samples' influence from a model – is critical for compliance, privacy, and model maintenance. Practically, retraining an $N$-sample model from scratch after removing even one sample incurs prohibitive cost, motivating a flurry of approximate unlearning methods. (Shen et al., 2024b;a; Hatua et al., 2024; Bourtoule et al., 2020; Cao & Yang, 2015; Golatkar et al., 2020; Ginart et al., 2019; Golatkar et al., 2021; Graves et al., 2020; Sekhari et al., 2021). However, despite this rapid progress, most unlearning work remains empirical and ad-hoc. We lack a unifying theoretical framework to predict when and why a given model can be efficiently unlearned. Notably, even initial theoretical treatments (e.g. from a differential-privacy viewpoint providing deletion guarantees (Sekhari et al., 2021; Chien et al., 2025)) do not explain the dynamics of forgetting or the interaction between the forget and retain sets. This gap motivates our work: we seek a principled understanding of the optimization dynamics of unlearning, grounded in the geometry of the model's loss landscape.

**Our approach.** We frame the unlearning process through the lens of asymptotic linear stability analysis in optimization. In this work, we analyze the asymptotic behavior of the stability of an unlearning solution, which fundamentally differs from the fine-tuning perspective (Ding et al., 2025) typically adopted in unlearning, and show that this stability is what ultimately determines whether a solution is unlearnable or not. Intuitively, unlike standard training which begins at random initialization, unlearning starts from a pre-trained model near a local minimum of the loss. We analyze small perturbation dynamics around that optimum to determine whether a "forgetting update" (e.g. gradient steps that increase loss on the forget set) will remain confined to a neighborhood of the original minimum (stable minima, no unlearning) or cause the model to drift off and diverge (unstable minima, unlearning possible). This linear stability perspective, inspired by prior analyses of SGD near minima (Ma & Ying, 2021; Dexter et al., 2024), provides a tractable characterization of the

transition between convergence vs. catastrophic forgetting. The key quantities in our analysis are a set of data coherence measures that quantify the alignment of loss gradients across samples near the optimum. We formally decompose coherence along three axes – (i) within the retain set, (ii) within the forget set, and (iii) between retain and forget sets – and derive stability thresholds in terms of these coherence values. Our theory thus for the first time links interactions between retain and forget data to unlearning success: for example, if the gradient directions of forget-set samples are highly aligned with those of retain-set samples (high retain-forget coherence), the model will resist unlearning because any parameter change that increases forget-set loss will also significantly hurt retain-set loss. Conversely, if the forget-set gradients live in a subspace largely independent from the retain-set (low inter-coherence), we prove the existence of a stable update direction that forgets the target data while leaving the rest of the model performance intact. These results yield a stability frontier in terms of data coherence: a boundary in data-geometry space separating regimes where gradient-based unlearning can succeed from where it fails.

Our framework also yields an intriguing insight into the relationship between training memorization and subsequent forgettability. Interestingly and perhaps surprisingly, we find that stronger memorization can make forgetting easier. In our framework, memorization corresponds to a regime of complex data where the model fits idiosyncratic details. We formalize this by adapting a two-layer ReLU CNN signal-plus-noise model from prior work on benign overfitting (Kou et al., 2023). Using random matrix theory tools, we prove that when the signal-to-noise ratio (SNR) in the data is lower (i.e. the model has to memorize more spurious noise), the cross-sample alignment of gradients is weaker – reducing the coherence terms, allowing effective forgetting of those samples. In contrast, a model trained on high-SNR data (with strongly aligned, dominant features) has very coherent gradients that push it to the edge of the stability frontier, making it resist unlearning – any attempt to forget one sample's influence will strongly perturb many others. We analytically identify this coherence-controlled stability boundary, and our experimental results confirm the trend: e.g. Hessian eigenvalue tests and CNN heatmaps of forget vs. retain influence align with the predicted boundary, mapping out how changes in batch size, mixing of forget/retain data, or network alignment affect the outcome of unlearning.

**Contributions** To summarize, our contributions are as follows: (1) We develop the first theoretical framework for machine unlearning based on linear stability analysis to address *what local optimization dynamics govern unlearning?*. We derive precise conditions (in terms of Hessian spectra and data coherence) under which standard gradient-based unlearning converges/diverges. (2) We address *how do the retain and forget sets interact, quantitatively, in determining stability?* Towards this goal, we introduce novel coherence metrics to quantify the retain–forget interaction, and prove how each coherence component (retain-retain, forget-forget, retain-forget) influences the stability of the unlearning process. These results formally characterize the joint role of data geometry and data distribution in forgetting dynamics. (3) To address *how does a model's propensity to memorize interact with its ability to forget?*, we establish a surprising link between memorization and forgettability: using a two-layer CNN with controllable noise, we prove that increased memorization (lower SNR) expands the range of stable unlearning (making forgetting easier), whereas high SNR (less overfitting) shrinks it (more resistant to forgetting). This is, to our knowledge, the first result to rigorously connect a model's generalization/memorization properties to its unlearning behavior. (4) Our empirical tests measure stability indicators (e.g. sharpness via Hessian eigenvalues) and unlearning performance under various conditions, and show strong agreement with the theoretical stability frontier. Taken together, our results provide the first principled account of the trade-offs between memorization, data coherence, and unlearning in modern ML models.

## 2 RELATED WORK

**Linear stability** Prior works (Wu et al., 2022; 2018; Wu & Su, 2023) utilize the linear stability framework to understand the relation between converging-diverging boundary and alignment of noise and loss landscape. Furthermore, Wu et al. (2022); Wu & Su (2023) connect the alignment properties to the simplicity bias that occurs in generic SGD. Additionally, Ma & Ying (2021) extend the framework by incorporating higher-order moments of the noise, revealing subtle implicit regularization effects on parameter evolution. Dexter et al. (2024) introduced the notion of data coherence, which directly quantifies the alignment of sample-specific gradients in the loss landscape, offering a fine-grained tool to analyze sample interactions. Compared to alternative theoretical

approaches, such as gradient flow or dynamical system approximations, linear stability has the distinct advantage of making explicit connections between model architecture, data distribution, and optimization algorithm. Unlike the standard learning scenario, unlearning requires analyzing the interleaving interaction between retain and forget sets, which introduces new dynamics not present in classical stability analysis. To address these challenges, we introduce new analytical tools and definitions that generalize coherence to mixed retain–forget settings. In doing so, we not only provide stability criteria for unlearning but also establish the first formal connection between memorization and forgetting, thereby broadening the scope of linear stability analysis beyond its traditional application to standard training.

**Theoretical works on Unlearning.** A number of foundational studies analyze machine unlearning by examining the optimization trajectory and its deviation from the original training dynamics. For example, Golatkar et al. (2020) model unlearning under a quadratic loss and characterize the drift in optimization trajectories by comparing the original and unlearned weights in the infinite-time limit. Ding et al. (2025) study approximate unlearning in linear models via weight-space distances, using the loss difference between the fine-tuned and unlearned models as the central metric. The recent work (Mavrothalassitis et al., 2025) analyzes linear logistic regression and expresses forgetting through closed-form weight difference between the original optimum and the unlearned solution. Thudi et al. (2022) unroll the SGD recursion to study unlearning dynamics through linearized gradient-flow approximation, and define unlearning directly in weight space, linking it to membership inference vulnerability. For further discussion, please see Appendix 5.4 and Appendix 5.1. Our work builds upon this prior theoretical foundation but departs conceptually: rather than characterizing forgetting via distance between solutions, we introduce a stability-based perspective grounded in optimization dynamics. By analyzing asymptotic linear stability of the unlearning update operator, we show how interactions between the retain and forget sets—mediated through curvature and data coherence—govern whether the optimizer will remain near the original minimum or diverge away, thereby determining whether unlearning is feasible.

## 3 Theory

### 3.1 Background

**Linear stability around minima.** Linear stability provides a principled lens for analyzing the local dynamics of iterative optimization near a critical point (e.g., local minima or saddles) by linearizing the update map and studying the resulting linear time-varying system to characterize convergence/divergence behavior of stochastic iterative algorithms for that critical point. This perspective underlies modern convergence analyses of SGD and its variants and, more recently, has proved effective for characterizing generalization-relevant phenomena such as rapid escape from sharp minima (Wu et al., 2018; Dexter et al., 2024). In our context, we consider a loss $L(w) = \frac{1}{n} \sum_{i=1}^n \ell_i(w)$ for model parameters $w \in \mathbb{R}^d$. Let $w^*$ be a local minimum. For a small perturbation $\delta$ around $w^*$, a first-order Taylor expansion gives

$$\nabla L(w^* + \delta) \approx \nabla^2 L(w^*)\, \delta,$$

since $\nabla L(w^*) = 0$. This linearization suggests that near $w^*$, the gradient is approximately given by the Hessian $H = \nabla^2 L(w^*)$ times the perturbation. Since we are only interested in the dynamics of the optimizer (rather than its absolute position), without loss of generality we take $w^* = 0$ as in prior works. For more discussion regarding assumption used in our work, please refer appendix 5.2.

**Stochastic gradient updates.** We are interested in the dynamics of stochastic gradient descent (SGD) near $w^*$. A generic SGD update can be written as

$$w_{t+1} = w_t - \eta\, g_t,$$

where $\eta > 0$ is the learning rate and $g_t$ is the stochastic gradient at step $t$. In the neighborhood of $w^*$, using the linear approximation, we can write $g_t \approx H_t w_t$, where $H_t$ is a random Hessian matrix (a mini-batch estimate of $H$). Thus the update becomes

$$w_{t+1} = w_t - \eta\, H_t w_t = (I - \eta H_t)\, w_t. \tag{1}$$

Here $H_t = \frac{1}{B} \sum_{i \in D_t} H_i$ is the average Hessian over the mini-batch $D_t$ of size $B$, and $H_i = \nabla^2 \ell_i(w^*)$. By construction $H = \frac{1}{n} \sum_{i=1}^n H_i$. Following Dexter et al. (2024), we model mini-batch sampling via Bernoulli selection (each data point is included in the batch independently with probability $B/n$).

**Sample wise gradient.** In the above equation, we assume that sample wise gradient at $w^*$ to be zero i.e.,

$$\nabla l_i(w) = \nabla l_i(w^*) + H_i(w - w^*).$$
$$= H_i(w - w^*). \tag{2}$$

This means that all gradients are results from curvatures. This assumption follows the standard linear interpolating regime used in prior works such as (Dexter et al., 2024; Wu & Su, 2023) and verified through empirical and theoretical studies (Tang et al., 2023; Chizat & Bach, 2020). In this regime, the local dynamics are dominated by the Hessian curvature to align with our focus. For more discussion regarding assumption used and limitation in our work, please refer appendix 5.2.

**Unlearning update rule.** To analyze machine unlearning (where a subset of the training data, called the *forget set*, is to be "forgotten" while the remaining data in the *retain set* is preserved), we adopt the update rule of Kurmanji et al. (2023). In this scheme, each step performs simultaneous gradient descent on the retain set and ascent on the forget set. Intuitively, this means we take a step that decreases loss on retained data while increasing loss on data that should be unlearned. Many gradient-based unlearning algorithms can be viewed as variants of this approach with different weighting of these components. We use $n_f, f_r$ for number of forget and retain samples respectively. In our linearized framework, the update with forget importance hyper-parameter $\alpha \in [0, 1]$ is:

$$w_{k+1} = w_k - \eta \Big[ (1-\alpha) \frac{1}{B} \sum_{i \in D_{r,k}} H_i \, w_k \; - \; \alpha \frac{1}{B} \sum_{i \in D_{f,k}} H_i \, w_k \Big], \tag{3}$$

where $D_{r,k}$ and $D_{f,k}$ denote the mini-batch of retain-set and forget-set examples at step $k$, respectively. This can be rewritten in operator form as

$$w_{k+1} = J_k \, w_k, \qquad J_k \; = \; I \; - \; \eta(1-\alpha) \frac{1}{B} \sum_{i \in D_{r,k}} H_i \; + \; \eta\alpha \frac{1}{B} \sum_{i \in D_{f,k}} H_i \,. \tag{4}$$

The random linear operator $J_k$ captures the combined effect of the retain and forget gradients at step $k$. Note that $J_k$ is itself random due to sampling of a mini-batch from each set.

A central question in linear stability analysis is whether the iterates remain near the original optimum or diverge away. To quantify this, we examine the expected squared norm of the parameters after $k$ steps, $\mathbb{E}\|w_k\|^2 = \mathbb{E}[w_k^T w_k]$. Starting from an isotropic small perturbation $w_0$ (we assume $w_0 \sim \mathcal{N}(0, I)$ without loss of generality), one can expand $w_k = J_{k-1} \cdots J_0 \, w_0$. This yields

$$\mathbb{E}\|w_k\|^2 \; = \; \mathbb{E}[w_0^T (J_0^T \cdots J_{k-1}^T J_{k-1} \cdots J_0) w_0] \; = \; \mathbb{E} \operatorname{Tr}\big( J_{k-1} \cdots J_0 J_0^T \cdots J_{k-1}^T \big), \tag{5}$$

where we used $\mathbb{E}[w_0 w_0^T] = I$ in the final equality. Eq (5) is the key quantity we will analyze to determine stability: if $\mathbb{E}\|w_k\|^2$ remains bounded (or decays) as $k \to \infty$, the unlearning process is *stable* (convergent) around $w^*$, whereas if $\mathbb{E}\|w_k\|^2 \to \infty$, the process is *unstable* (diverges, escaping $w^*$). For more discussion regarding the assumptions used in our work, please refer Appendix 5.2, and for more discussion regarding divergence and unlearning, please refer to Appendix 5.4.

### 3.2 Coherence Measures for Unlearning

**Coherence in single-dataset SGD.** Before introducing our new coherence measures tailored to unlearning, we briefly review the original notion of *Hessian coherence* from Dexter et al. (2024) for standard (single dataset) learning. The coherence quantifies the alignment between per-sample Hessians in the training set. Intuitively, if all samples induce very aligned curvature directions, SGD will experience less "randomness" and more stable updates, whereas if each sample's loss landscape is oriented differently, the optimization dynamics are more erratic.

**Definition 1** (Coherence, single set (Dexter et al., 2024)). *Given a collection of positive semidefinite (PSD) Hessian matrices $\{H_i : i \in [n]\}$ for $n$ training samples, the* coherence matrix $S \in \mathbb{R}^{n \times n}$ *is defined by*

$$S_{ij}^{single} \; = \; \| H_i^{1/2} H_j^{1/2} \|_F \,,$$

*the Frobenius norm of the product of the square-root Hessians of sample $i$ and $j$. The associated coherence measure is*

$$\sigma^{single} \; = \; \frac{\lambda_{\max}(S^{single})}{\max_{i \in [n]} \lambda_{\max}(H_i)} \,,$$

*i.e. the largest eigenvalue of $S^{single}$ normalized by the largest individual sample Hessian eigenvalue.*

Intuitively, $\sigma^{\text{single}}$ close to 1 indicates that the top curvature directions of all samples are closely aligned (high coherence), whereas a small $\sigma^{\text{single}}$ indicates disparate or orthogonal curvatures across samples. Prior work showed that higher coherence $\sigma^{\text{single}}$ correlates with greater stability of SGD. In other words, when the loss landscapes of different samples "point" in similar directions, gradient steps reinforce each other and it is more resistant for SGD to diverge away from the optimum. For discussion about the practical feasibility of calculation for coherence matrix, please refer to Appendix 5.5.

**Coherence with retain and forget sets.** In an unlearning scenario, we have two disjoint sets of samples: the retain set $D_r$ (of size $n_r$) and the forget set $D_f$ (of size $n_f$). Coherence within each set (retain vs. forget) is not sufficient to describe the behavior of the combined ascent-descent dynamics. We need to also quantify the interaction *between* the two sets. We therefore introduce a series of definitions that extend coherence to the multi-set setting.

First, we define a weighted combination of Hessians from a retain-forget pair, which will serve as an effective "mixing Hessian:"

**Definition 2** (Mix-Hessian).

$$D := \frac{1}{n_r n_f} \sum_{r \in D_r, f \in D_f} \frac{C_r^{\frac{1}{2}}}{C_r^{\frac{1}{2}} + C_f^{\frac{1}{2}}} H_r + \frac{C_f^{\frac{1}{2}}}{C_r^{\frac{1}{2}} + C_f^{\frac{1}{2}}} H_f = \frac{1}{n_r n_f} \sum_{rf} D_{rf}, \tag{6}$$

Here $C_r = \eta^2(1-\alpha)^2 \frac{1}{n_r}(\frac{1}{B} - \frac{1}{n_r})$, $C_f = \eta^2 \alpha^2 \frac{1}{n_f}(\frac{1}{B} - \frac{1}{n_f})$, $D_{rf} = \frac{C_r^{\frac{1}{2}}}{C_r^{\frac{1}{2}} + C_f^{\frac{1}{2}}} H_r + \frac{C_f^{\frac{1}{2}}}{C_r^{\frac{1}{2}} + C_f^{\frac{1}{2}}} H_f$. The constants $C_r$ and $C_f$ reflect the relative contribution of retain vs forget Hessians to the second-moment dynamics of $w_k$ (these arise from the SGD noise analysis in Lemma 3.1 later). The mix-Hessian $D$ aggregates the pairwise influence of retain and forget sets; it effectively summarizes how the two sets jointly affect curvature when considered together in the update. Next, analogous to the single-set case, we define a coherence matrix that captures alignment across *pairs* of retain/forget examples:

**Definition 3** (Mix-coherence matrix). *Construct an index set for all retain–forget pairs: $\mathcal{P} = \{(r, f) : r \in D_r, f \in D_f\}$ of size $|\mathcal{P}| = n_r n_f$. The* mix-coherence matrix $S \in \mathbb{R}^{|\mathcal{P}| \times |\mathcal{P}|}$ *is defined entrywise by*

$$S_{(r,f),(r',f')} = \| D_{rf}^{1/2} D_{r'f'}^{1/2} \|_F,$$

*for any $(r, f), (r', f') \in \mathcal{P}$.*

In words, $S$ measures the alignment between every pair of mixed Hessians $D_{rf}$ and $D_{r'f'}$. Finally, we define an overall coherence measure for unlearning, generalizing the single-set $\sigma$:

**Definition 4** (Unlearning Coherence Measure). *The unlearning coherence is*

$$\sigma = \frac{\lambda_{\max}(S)}{\max_{(r,f) \in \mathcal{P}} \lambda_{\max}(D_{rf})},$$

*the leading eigenvalue of the mix-coherence matrix $S$ normalized by the largest eigenvalue among all individual $D_{rf}$ matrices.*

This definition reduces to the original coherence measure in the limit where one of the sets is absent (e.g. $n_f = 0$ or $\alpha = 0$ yields only a retain set). It simultaneously captures the within-set coherence and the cross-set coupling. Intuitively, if the retain and forget sets are highly *aligned* in terms of curvature directions, the mix-coherence $\sigma$ will be large. In that case, performing ascent on forget and descent on retain will tend to cancel out: the update directions from the two sets are similar but with opposite sign, leading to minimal movement away from $w^*$. This predicts stability for the current optimum (i.e. resistance to unlearning). Conversely, if the two sets are incoherent (small $\sigma$), their Hessians push in different directions; the ascent on forget set will not be canceled by descent on retain, making it easier for the iterates $w_k$ to escape the original minimum. In summary, our multi-set coherence measure $\sigma$ quantifies how conducive the data geometry is to either divergence or convergence during unlearning. To our knowledge, this is the first work to explicitly incorporate multiple data subsets into a stability analysis of optimization.

## 3.3 Linear Stability Analysis of Unlearning

In the theory section, we study a more fundamental problem regarding whether a set is unlearnable in asymptotic manner through optimization instead of in a fine-tuning regime. We now leverage the above framework to derive conditions under which the unlearning dynamics (3) will converge or diverge. In our work, A key technical challenge is that, unlike in the single-set case, the influence of SGD noise cannot be captured by a simple closed-form recursion. This is because the gradient noise now comes from two interleaved sources (retain and forget sets) with potentially different magnitudes.

We begin with a lemma that describes the evolution of the second moment $\mathbb{E}\|w_k\|^2$ in terms of a recursive sequence of matrices $N_k$. This lemma generalizes the stability condition from Dexter et al. (2024) to account for the alternating ascent/descent updates.

**Lemma 3.1** (Stability recurrence for unlearning). *Consider the unlearning update operator $J_k$ defined in* (4). *Define a sequence of PSD matrices $\{N_k\}_{k\geq 0}$ by $N_0 = I$ and for $k \geq 1$:*

$$N_k = C_f \sum_{i \in D_f} H_i N_{k-1} H_i + C_r \sum_{i \in D_r} H_i N_{k-1} H_i, \tag{7}$$

*with $C_r, C_f$ as given in Definition 2. Also let $M_k = J^{2k} + N_k$. ($J = I - \eta(1-\alpha)H_R + \eta\alpha H_F$ where $H_R$ and $H_F$ ar full Hessian of retain and forget set. See definition 6) Then:*

1. *(**Lower bound**) $\mathbb{E}\,\mathrm{Tr}\big(J_0^T \cdots J_{k-1}^T J_{k-1} \cdots J_0\big) \geq \mathrm{Tr}(M_k)$. Moreover, if $\mathrm{Tr}(N_k) \to \infty$ as $k \to \infty$, then $\mathbb{E}\|w_k\|^2 \to \infty$ as well.*

2. *(**Upper bound**) If at each step $J_k$ is spectrally bounded as $(1-\epsilon)I \succeq J \succeq -(1-\epsilon)I$ for some $\epsilon \in (0,1)$ (i.e. all eigenvalues of $J$ lie in $[-(1-\epsilon), 1-\epsilon]$), then*

$$\mathbb{E}\,\mathrm{Tr}\big(J_0^T \cdots J_{k-1}^T J_{k-1} \cdots J_0\big) \leq \sum_{r=0}^{k-1} \binom{k}{r}(1-\epsilon)^{2(k-r)}\,\mathrm{Tr}(N_r).$$

   *If in addition $\mathrm{Tr}(N_r) \leq \epsilon$ for all $r$, then $\mathbb{E}\|w_k\|^2 \to 0$ as $k \to \infty$ (the unlearning update converges in mean square).*

*Discussion.* Part (1) of Lemma 3.1 provides a sufficient condition for divergence: if the "noise accumulation" matrices $N_k$ (which capture how SGD variance builds up over iterations) have unbounded trace, then the model will eventually blow up (escape the optimum). Part (2) gives a sufficient condition for convergence: if each $J$ is a contraction (spectral norm < 1 by a margin $\epsilon$) and the accumulated noise remains small, then the model's parameter norm will vanish (meaning $w_k$ returns to the optimum). These statements generalize classical stability results to the unlearning case. Importantly, the recursion (7) for $N_k$ does not admit a simple closed form because $N_{k-1}$ appears inside sums over both sets $D_r$ and $D_f$. This coupling between retain and forget sets is what makes analyzing unlearning challenging. By introducing the coherence measures (Definition 4 and related definition), we overcome this hurdle: the coherence will allow us to relate $\mathrm{Tr}(N_k)$ to data-dependent quantities like $\lambda_{\max}(D)$ and thereby derive interpretable stability criteria.

Using the coherence framework, we can now state our main stability thresholds. The first result is a condition under which the unlearning dynamics *diverge* (fail to stay at the original minimum):

**Theorem 3.2** (Divergence criterion for unlearning). *Under the setup of Lemma 3.1, the unlearning process will diverge if the mix-Hessian eigenvalue exceeds a threshold determined by the coherence. In particular, if*

$$\lambda_{\max}(D) \geq \frac{\sqrt{2}\,\sigma}{\eta\Big((1-\alpha)\,n_f\,\sqrt{\frac{n_r}{B}-1} + \alpha\,n_r\,\sqrt{\frac{n_f}{B}-1}\Big)}, \tag{8}$$

*then $\lim_{k \to \infty} \mathbb{E}\|w_k\|^2 = \infty$. Equivalently, condition* (8) *guarantees the unlearning algorithm will escape the original minima (diverge) due to the stochastic dynamics.*

In plain terms, Theorem 3.2 says that if the influence of the forget–retain interaction (measured by $\lambda_{\max}(D)$) is sufficiently large relative to the stabilizing effect of coherence $\sigma$ (and other factors like

batch size $B$ and relative set sizes), then the gradient ascent on the forget set will overpower the descent on the retain set, leading to instability. The inequality (8) can be viewed as a quantitative stability limit or "edge of chaos" for unlearning: beyond this point, the original solution $w^*$ cannot hold.

Our next theorem establishes a matching lower bound, showing that the above divergence condition is essentially tight. It guarantees that when $\lambda_{\max}(D)$ is below a certain threshold (of the same order as in (8)), one can find a scenario where the unlearning process converges, thereby demonstrating that the threshold cannot be significantly improved in general:

**Theorem 3.3** (Convergence condition (matching lower bound))**.** *Suppose* $\lambda_{\max}(D)$ *and* $\sigma$ *satisfy*

$$\lambda_{\max}(D) \ \leq \ \frac{2\,\sigma}{\eta\,C_r'\left(\sigma + n_f\left(\frac{n_r}{B} - 1\right)\right)}\,, \tag{9}$$

*where* $C_r' = \sqrt{C_r}/(\sqrt{C_r} + \sqrt{C_f})$ *(with* $C_r, C_f$ *from Definition 2). Then there exists a choice of PSD Hessians* $\{H_i\}$ *for the retain and forget sets such that the unlearning update converges (i.e.* $\lim_{k \to \infty} \mathbb{E}\|w_k\|^2 = 0$) *under those Hessians.*

The convergence condition (9) mirrors the divergence condition in its dependence on $\sigma$, $n_r, n_f$, and $B$. The existence of a construction that achieves convergence when (9) holds indicates that our divergence criterion in Theorem 3.2 is tight up to constant factors. In summary, Theorems 3.2 and 3.3 together pin down a theoretical threshold curve in the space of data coherence and algorithm parameters that separates stable (convergent) unlearning from unstable (divergent) unlearning. We can now interpret some common unlearning strategies:

**Naive negative gradient.** A straightforward unlearning baseline is to set $\alpha = 1$ and run gradient ascent on the forget set alone. Our framework explains why this often fails. If the forget set has high internal coherence, its gradients align with the curvature at $w^*$, so ascent follows a single stable direction and does not escape the minimum due to lack of stochasticity. Without rendering of stochasticity, the small learning rate can give slow diverging behavior. If the forget and retain sets are also highly coherent, the overall coherence stays large even without the retain set. In both cases divergence is inhibited or slowed down, matching empirical reports that naive negative-gradient unlearning typically stagnates or oscillates, hurting retained data while barely reducing forget-set performance (Ding et al., 2025; Fan et al., 2025; Ding et al., 2025).

**Random label perturbation.** Another strategy is to add randomness to the forgetting process, for instance by using mislabeled data or injecting noise into the forget set's gradients (see, e.g., random label unlearning). In our terms, this deliberately *breaks the coherence* of the forget set: if labels are randomized, the gradients from forget-set samples become effectively uncorrelated, dramatically lowering the forget-set's internal $\sigma$. This, in turn, allows the model to escape the original minimum much faster. Moreover, randomizing forget labels also reduces the coupling between forget and retain sets (since the forget-set gradient is now essentially random noise orthogonal to the retain-set Hessians). Thus, random label methods improve unlearning by driving the coherence measure $\sigma$ downward, so the divergence criterion is more easily satisfied. (Graves et al., 2020)

**Min-Max (targeted forget) methods.** More sophisticated approaches pick a subset of model weights or directions that are most "responsible" for the forget set's performance, and then apply ascent/descent on those components. This can be seen as applying projection matrices $P_F$ and $P_R$ to the Hessians $H_f, H_r$ respectively, focusing updates on certain eigen-directions. Such projections effectively reduce the overlap between forget-set and retain-set update directions (since $P_F H_f$ and $P_R H_r$ act in different subspaces), thereby reducing the cross-coherence between the two sets. In our framework, this corresponds to a smaller overall $\sigma$ as well. By isolating the forgetting dynamics, Min-Max methods thus decrease the ability of the retain set to interfere with forgetting (and vice versa), making the unlearning process more effective. (Tang & Khanna, 2025; Fan et al., 2024)

### 3.4 MEMORIZATION AND FORGETTING

So far, our analysis has focused on the role of stochastic gradient noise (from mini-batch sampling). We now turn to another key factor: the inherent *signal vs. noise* structure of the data itself. We ask: if a model has *memorized* certain training examples (as opposed to learning a shared signal), does that make it easier or more resistant to forget those examples? We will show a theoretical connection

between a model's tendency to memorize (which occurs when data has low signal-to-noise ratio) and the ease of unlearning. Our work aims to identify the memorization resulting from highly orthogonal component (noise, outlier features) and its relationship to forgetting (see Appendix 5.6 for further discussion.) To make this concrete, we consider a specific data model and network, inspired by the theoretical construction by Kou et al. (2023). The data distribution is designed so that each example contains a mixture of a common signal and independent noise. This is formalized as follows:

**Definition 5** (Data Setup). *Let $\mu \in \mathbb{R}^d$ be a fixed unit-norm signal vector. Each training example consists of a feature pair $x = [x^{(1)}; x^{(2)}] \in \mathbb{R}^{2d}$ (concatenation of two d-dimensional parts) and a label $y \in \{-1, +1\}$. The example is generated by:*

1. *Sample $y$ as a Rademacher random variable ($\Pr(y = +1) = \Pr(y = -1) = \frac{1}{2}$).*

2. *Sample a noise vector $\xi \sim \mathcal{N}(0, \sigma^2 I_d)$ in $\mathbb{R}^d$, where $\sigma^2$ is the noise variance.*

3. *With equal probability, set either $x^{(1)} = y\,\mu$ and $x^{(2)} = \xi$, or $x^{(1)} = \xi$ and $x^{(2)} = y\,\mu$. In other words, one of the two halves of $x$ carries the signal $y\,\mu$ and the other carries independent noise.*

We then consider a two-layer convolutional neural network (CNN) with ReLU activations operating on this data.. The network has two sets of convolutional filters (for the positive and negative class) and outputs a score $f(W, x)$ whose sign determines the predicted label. Specifically, let $W^{(+1)}$ and $W^{(-1)}$ be the weight matrices for the two classes, each of shape $m \times d$ (with $m$ filters). The network's output is

$$
\begin{aligned}
f(W, x) &= \frac{1}{m} \sum_{r=1}^{m} \Big( \text{ReLU}(\langle w_r^{(+1)}, x^{(1)} \rangle) + \text{ReLU}(\langle w_r^{(+1)}, x^{(2)} \rangle) \Big) \\
&\quad - \frac{1}{m} \sum_{r=1}^{m} \Big( \text{ReLU}(\langle w_r^{(-1)}, x^{(1)} \rangle) + \text{ReLU}(\langle w_r^{(-1)}, x^{(2)} \rangle) \Big),
\end{aligned}
\tag{10}
$$

and the model is trained with logistic loss $L_S(W) = \frac{1}{n} \sum_{i=1}^{n} \log\big(1 + \exp(-y_i\,f(W, x_i))\big)$. We focus on the case where the network can fit the training data perfectly (interpolating regime) and potentially overfits.

In this setting, we can analyze the coherence of the Hessians at the trained solution. The following result provides an upper bound on the coherence in terms of the *signal-to-noise ratio* (SNR) of the data, defined as $\text{SNR} = \frac{\|\mu\|}{\sigma\sqrt{d}}$ (which measures the strength of the common signal relative to noise in each example):

**Theorem 3.4** (Coherence bound in the CNN memorization model). *Under the data model of Definition 5 and the two-layer ReLU CNN defined above, suppose the network is trained to near-zero training loss. Then with probability at least $1 - 8\delta$ (over the random draw of the dataset), the largest eigenvalue of the coherence matrix S for the retain/forget split satisfies*

$$
\lambda_{\max}(S) \leq \mathcal{O}\Big( n_r\, n_f\, d\sigma^2 \Big[ (\sqrt{C'_r} + \sqrt{C'_f})^2\, (\text{SNR})^2 + (C'_r + C'_f) \Big] \Big),
\tag{11}
$$

$$
\max_{rf} \lambda_{\max}(D_{rf}) \leq \mathcal{O}\big( (C'_r + C'_f)(d\sigma^2(\text{SNR})^2 + 1) \big),
\tag{12}
$$

*where $C'_r$ and $C'_f$ are the normalized retain/forget weight fractions as defined in Theorem 3.3. Consider division of two quantities and we can find that for small SNR limit and large SNR limit:*

$$
\lim_{\text{SNR} \to 0} \frac{\lambda_{\max}(S)^{upper}}{\max_{rf} D_{rf}^{upper}} = \mathcal{O}(n_r n_f) \;,\; \lim_{\text{SNR} \to \infty} \frac{\lambda_{\max}(S)^{upper}}{\max_{rf} D_{rf}^{upper}} = \mathcal{O}\Big(n_r n_f \big(1 + \frac{2\sqrt{C'_r C'_f}}{C'_r + C'_f}\big)\Big).
\tag{13}
$$

**Discussion** Theorem 3.4 shows the surprising role of SNR in stability of the optimizer through its control over the coherence. In particular, if the data has a very low SNR (meaning $\mu$ is small relative to the noise $\sigma$), then the network is likely to memorize the noise. In that regime, high-dimensional random noise vectors are nearly orthogonal to each other, so Hessians for different samples align poorly. Our bound indicates that coherence measure is larger in large SNR limit compared to

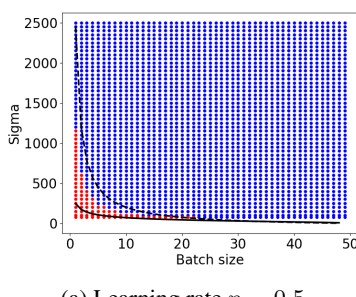
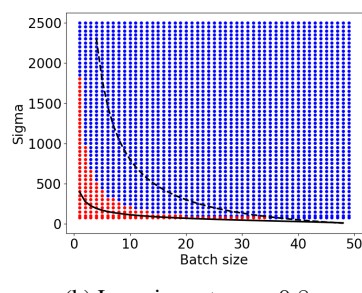

(a) Learning rate $\eta = 0.5$        (b) Learning rate $\eta = 0.8$

Figure 1: **Tight upper and lower bounds.** Blue = convergence, red = divergence. The dashed line is the lower bound (Theorem 3.3), the solid line the divergence criterion (Theorem 3.2). Both closely track the true boundary.

small SNR limit, so a smaller SNR yields a smaller coherence. Consequently, when the model has memorized (low SNR), the unlearning process becomes easier: the model can move away from the original fit with less resistance, as formalized by our earlier stability criteria. Conversely, if the data has high SNR (dominant signal shared across examples), the model will latch onto that signal, resulting in large coherence, and the the model resists leaving the optimum since all samples agree on the direction.

This gives a rigorous basis for a perhaps counter-intuitive aphorism: *the more you memorize, the easier you forget.* In other words, models that rely heavily on idiosyncratic features of individual data points (memorization) are in fact less stable at those points and can forget them with less effort, whereas models that have learned a strong global structure (signal) are more stable and resistant to having a single sample's influence removed. Our work is the first to formally establish this connection between memorization (in terms of data geometry) and unlearning. We believe this provides valuable insight into the trade-offs inherent in machine unlearning.

## 4 EXPERIMENTS

### 4.1 DIVERGING AND CONVERGING CONDITION.

**Experimental setup.** In this section, we simulate experiments to test Theorems 3.2 and 3.3. We fix $n_f = n_r = 50$ and set $\alpha = 0.1$. Say $Q$ is a hyper-parameter constant. We will set $Q$ to different values to control various quantities in the experiments. For the retain set, Hessians are defined as $H_i = m e_1 e_1^T$ for $i \in [Q]$, and $H_i = m e_{i-Q+1} e_{i-Q+1}^T$ otherwise, with $m = 2n_r/Q$; the forget set uses the same construction. This ensures $\lambda_1(H_R) = \lambda_1(H_F) = \lambda_1(D) = 2$, controlling sharpness. We choose $\eta \leq 1$ to avoid divergence from the standard criterion $\eta \geq 2/\lambda_1$, so any escaping behavior stems solely from stochasticity, consistent with our theorem.

To vary coherence, we change $Q$ and compute $(B, \sigma)$ pairs by adjusting batch size. For each pair, we randomly initialize $w$, run 1000 updates, and record $\|w_{1000}\|$. Runs with $\|w_{1000}\|/\|w_0\| \geq 1000$ are marked as diverging. Each experiment is repeated 10 times, and the majority outcome determines convergence/divergence.

**Bounds on divergence.** Figure 1 shows that both our upper and lower bounds predict the divergence region accurately; the bounds are tighter for batch sizes $\geq 10$. The divergence criterion in particular matches the true boundary, demonstrating that our coherence-based measure captures the essential optimization dynamics accurately. This highlights coherence as a meaningful lens on unlearning dynamics, with potential applications beyond our scope. Please see appendix 5.3 for further details.

### 4.2 RELATION BETWEEN MEMORIZATION AND FORGETTING.

**Experimental setup.** We generate data as in Definition 5 along with the 2 layer CNN. The dataset has 50 training samples without label noise. We set $\boldsymbol{\mu} = \|\boldsymbol{\mu}\|_2 [1, 0, \ldots, 0]$ and add Gaussian noise $\xi \sim \mathcal{N}(0, \sigma^2 I_d)$ with $\sigma = 1.0$. To control the SNR in our experiments, we vary the signal strength to

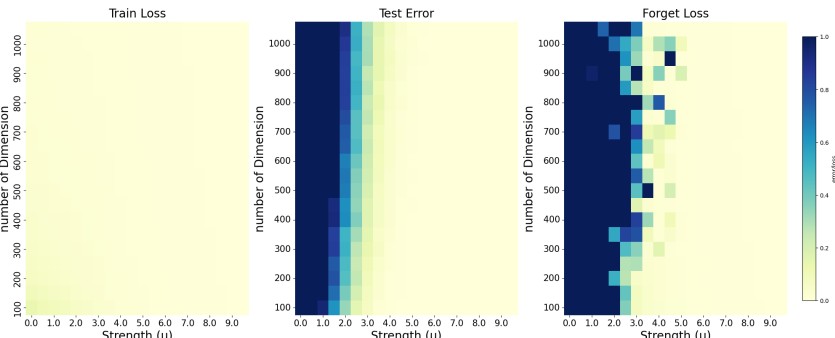

Figure 2: **(Left)** Training loss. **(Middle)** Test error. **(Right)** Forget loss. Memorization and forgetting regions strongly overlap as indicated by the overlap blue region.

achieve different level of SNR while fixing the noise magnitude. We vary the value of $d \in [100, 1100]$ to verify our results with varying levels of over-parametrization. The CNN has $m = 10$ filters and is trained by full-batch gradient descent for 100 epochs at learning rate 0.1, ensuring training loss $\leq 0.1$.

We record training loss and test error (on 1000 unseen samples). For unlearning, out of total 50 samples, we use 25 samples to form the forget set and the other 25 to build the retain set. We apply mini-batch unlearning (batch size 5) using the negative-gradient method with learning rate 0.1 and $\alpha = 0.3$ for 90 steps, then record the average forget loss. Each experiment is repeated 20 times and averaged.

**Memorization–forgetting overlap.** Figure 2 shows heatmaps over signal strength and dimension. Memorization is identified where training loss is low but test error high (left, middle). Strikingly, these regions coincide with high loss on the forget (right), confirming our prediction: memorization corresponds to low coherence, making solutions unstable and easier to escape, thus making unlearning easier. This provides strong evidence for our framework and, to our knowledge, is the first work to connect memorization and forgetting through coherence. For more detailed discussion regarding the setup and its corresponding purpose, please refer appendix 5.3.

**Larger scale and real world data.** To validate our results on real world scenarios, we conduct additional experiments on a more realistic setting: CIFAR-10 with a ResNet-18 model. We first train the model to convergence (100 percent training accuracy). We then perform unlearning steps as stated in eq 3. We use step size as 0.01 with forget set being 10 percent of the training set. We set $\alpha$ or weighting between forget set and training set is set to 0.3. We record the loss on the forget set for the first 500 steps at interval of 50 steps. To probe the relationship between memorization and unlearning predicted by our theory, we inject Gaussian noise of varying variance (0.1,0.3,0.5) into the inputs. Higher noise variance produces stronger memorization, since the network overfits the idiosyncratic noise patterns. As shown in Table 1 and predicted by our coherence framework, models with higher memorization (larger variance) exhibit *faster loss increase on the forget set* during unlearning.

| Step | Var = 0.1 | Var = 0.3 | Var = 0.5 |
|---|---|---|---|
| 0 | $0.0016\pm0.0005$ | $0.0019\pm0.0004$ | $0.0016\pm0.0004$ |
| 50 | $0.0024\pm0.0014$ | $0.0016\pm0.0003$ | $0.0700\pm0.0730$ |
| 100 | $0.0694\pm0.0858$ | $0.0428\pm0.0539$ | $0.0715\pm0.0667$ |
| 150 | $0.0477\pm0.0508$ | $0.0356\pm0.0401$ | $0.1575\pm0.1692$ |
| 200 | $0.0722\pm0.0762$ | $0.1582\pm0.1744$ | $0.6279\pm0.5556$ |
| 250 | $0.1758\pm0.2054$ | $0.1671\pm0.1508$ | $1.1366\pm0.6084$ |
| 300 | $0.3888\pm0.4347$ | $0.5626\pm0.5598$ | $1.4868\pm0.9849$ |
| 350 | $0.6515\pm0.7707$ | $0.8544\pm0.7993$ | $2.4464\pm1.5124$ |
| 400 | $1.4362\pm1.5605$ | $2.3895\pm1.2796$ | $5.5148\pm1.8706$ |
| 450 | $2.5029\pm2.3457$ | $2.9324\pm1.2882$ | $5.5811\pm1.8397$ |

Table 1: Forget-set loss (mean $\pm$ std) during unlearning across noise levels.

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

# 5 APPENDIX

## 5.1 ADDITIONAL RELATED WORKS

**Machine unlearning and memorization.** Many unlearning methods are proposed to effectively erase information of selected samples. Several basic but well-known methods such as random labeling of forget set (Graves et al., 2020) and explicit gradient ascent on the forget set (Warnecke et al., 2023) lay foundation for current unlearning methods. More recent works extend on those works to improve overall performance of unlearning. For example, SCRUB (Kurmanji et al., 2023) simultaneously perform gradient ascent on forget set and gradient descent in the retain set to better preserve performance of retain during unlearning. Influence based (Izzo et al., 2021) unlearning propose idea that takes into account of the Hessian information of datasets to perform update of the model weights. Saliency Unlearning (Fan et al., 2024) identity weights that react strongly to forget set through magnitude of gradient and perform unlearning only on those weight to achieve better performance. There are several theoretical studies about unlearning through the lens of the differential privacy and provide performance guarantee. For example, Langevin Unlearning Chien et al. (2025) study unlearning with privacy guarantee through projected noisy gradient descent. Sekhari et al. (2021) studies unlearning problem and provide performance guarantee and the corresponding sample complexity. There are also works discussing relationship between memorization and generalization. Attias et al. (2024) discuss the fundamental trade-off between generalization and memorization under information theory framework. Carlini et al. (2019) discuss different metrics for identifying sample of different type (memorized, prototypical and so on). Feldman (2021) provide theoretical and experimental analysis saying the memorization is necessary to achieve optimal performance. There are also several works studying memorization with different tasks and model architectures (Biderman et al. (2023); Li et al. (2025); Prashanth et al. (2025)).

## 5.2 DISCUSSION ABOUT ASSUMPTIONS

**Linearized dynamics and quadratic approximation:** In this section, we provide a consolidated and detailed discussion for why this modeling choice is (1) standard, (2) empirically grounded, and (3) necessary for theoretical progress in unlearning.

First, the local quadratic approximation is a well-established and widely validated modeling framework in the theory of deep learning. A large body of recent work successfully explains diverse optimization behaviors using this approximation, including the Edge-of-Stability phenomenon (Andreyev & Beneventano, 2025; Lee & Jang, 2023), implicit bias and generalization (Wu & Su, 2023; Wu et al., 2022), eigenvalue dynamics and curvature structures (Agarwala & Dauphin, 2023), and stability versus divergence of SGD (Dexter et al., 2024; Chang & Khanna, 2025). While the approximation does impose limitations as (all theoretical frameworks do) it also provides meaningful insights into real-world systems that alternative approaches currently cannot offer. The success of these works illustrates that quadratic modeling is both theoretically fruitful and empirically relevant. Use of the same approximation follows this tradition and enables us to reveal new connections between the forget set and retain set via their geometric interactions.

Second, multiple empirical studies show that the loss landscape near well-trained minima is smooth and locally well-approximated by a quadratic form (Li et al., 2018; Singh & Alistarh, 2020; Ghorbani et al., 2019). This has been repeatedly observed across architectures, datasets, and training regimes. Importantly, unlearning is a fine-tuning procedure that begins from an already-converged minimum, precisely the region where such local approximations are the most accurate. Thus, the theoretical foundation is not only supported by prior empirical evidence but is also particularly appropriate for modeling unlearning dynamics.

Third, a key question is: why do we need approximations at all in theoretical unlearning? Existing theoretical works on machine unlearning also rely on simplified or linearized systems in order to obtain analyzable results. For example, Golatkar et al. (2020) study unlearning under a quadratic model and analyze the drift of optimization trajectories which is exactly the type of measure we employ. Ding et al. (2025) use linear feature–weight dot product models and weight-space distances to characterize approximate unlearning. The recent work (Mavrothalassitis et al., 2025) operates in linear logistic regression, enabling closed-form solutions and a theoretical characterization of

forgetting difficulty. Thudi et al. (2022) analyze unlearning by unrolling the SGD recursion, which again relies on linearization of the gradient dynamics.

One commonality across all these theoretical frameworks is that they require simplifying assumptions to make the problem analyzable. Our quadratic approximation fits within this established methodology. Although no single abstraction captures the full complexity of deep networks, each sheds light on a different aspect of the phenomenon. Our framework contributes by offering a principled geometric description of how the retain and forget sets interact through curvature and coherence, revealing a new mechanistic explanation for when gradient-based unlearning becomes difficult or feasible.

**Sample wise gradient** This assumption used in our work required that sample wise gradient at $w^*$ to be zero i.e.,

$$
\begin{aligned}
\nabla l_i(w) &= \nabla l_i(w^*) + H_i(w - w^*) \\
&= H_i(w - w^*)
\end{aligned}
\tag{14}
$$

This means that all gradients around the optimum strictly depend on curvatures. This aligns with the focus of our work which is to study how curvatures influence the optimization behavior. This assumption follows the standard linear interpolating regime used in prior works such as (Dexter et al., 2024; Wu & Su, 2023) widely observed in overparameterized networks, where training loss and sample-wise losses approach zero near convergence. Empirical and theoretical studies (Tang et al., 2023; Chizat & Bach, 2020) have shown that modern neural networks can fit all training labels exactly, implying that the close-to-zero loss results from vanishing per-sample gradients rather than balanced large gradients. In this regime, the local dynamics are dominated by the Hessian curvature, making the linear approximation around minima both standard and justified.

### 5.3 More discussion about the experiments

Below we clarify the design and purpose of our experiments and will add the discussion into the updated version of our work. The experiments serve two distinct goals: (1) verifying the theoretical divergence criterion and matching lower bound under a controlled synthetic setting, and (2) empirically testing the predicted link between memorization and forgetting using a two-layer CNN.

**Synthetic verification.** We construct sample-wise Hessians such that the overall dataset Hessian is constrained by $2 / \eta$, ensuring that any divergence behavior arises purely from stochasticity and data geometry. We vary the sample-wise Hessian coherence while maintaining the same global sharpness, then run training under different learning rates. Divergence is detected when the weight norm grows 1000× larger than initialization (repeated over five runs). As shown in Fig. 1, the empirically observed divergence boundary (red/blue transition) aligns closely with both our theoretical criterion and the matching lower bound, confirming that the theory precisely predicts optimization stability.

**Memorization–forgetting test.** For the second part, we want to verify the whether or not stronger memorization will lead to stronger forgetting or unlearning as our theory indicates that when model memorize the data, it will map the data to space where it is orthogonal to the main dataset in terms of loss curvatures and give low coherence measure. This low coherence measure will therefore lead to easier forgetting process as predicted by our theory. To control memorization strength, we generate datasets with varying signal-to-noise ratio (SNR). Low-SNR data force the model to memorize noise (learn orthogonal noise directions in the loss curvature space) yielding low coherence. After training, we apply our unlearning procedure and track forget losses. The training loss indicates that all models properly learn the data and converge. The testing loss is to demonstrate whether or not the model memorize the data. This is due to the fact that the when the model perform well in train but bad in the testing, it indicate there exist memorization. We therefore can observe that small SNR regime show strong memorization. Lastly, the forget loss indicate that whether or not we can escape the current minima and perform successful unlearning. Our results show that it is consistence with our theoretical prediction. The regime of memorization will also give stronger forgetting results aligning with our coherence measure.

## 5.4 Discussion about divergence and unlearning.

In this section, we discuss how the divergence and unlearning are related under different scopes of unlearning and how is this relationship being explored in prior works. Further more, we want to answer why the loss or distance based metric is still valuable and remain one stander for unlearning problem.

First, divergence and distance is widely used unlearning metrics in many prior works. while there is currently no single universal definition of unlearning across domains: some applications emphasize privacy (MIA resistance), others focus on confusion removal, bias removal, safety, or utility preservation. Despite this diversity, loss and distance based metrics remain among the most commonly used evaluation tools, and our work is consistent with this long line of literature. Our use of the metric is consistent with many established works that continue to evaluate unlearning effectiveness using forget-set loss or accuracy. Examples include unlearning accuracy (Fan et al., 2024), forgetting error (Kurmanji et al., 2023), forget performance (Kurmanji et al., 2023), forgetting loss (Graves et al., 2020; Golatkar et al., 2020) and distance-based measures (Thudi et al., 2022). They are also work characterizing the distance between solutions resulting from different algorithms (Mavrothalassitis et al., 2025). These metrics are still used because they capture the effect of removing the forget set and they correlate strongly with practical privacy risks (e.g., MIA success). In this sense, analyzing divergence or distance quantities that govern how the model leaves the old solution is aligned with standard practice.

Second, why divergence is theoretically meaningful and aligns with prior unlearning theory? Several foundational works analyze unlearning by studying optimization trajectory and its reaction to different component involved in unlearning. For example, Golatkar et al. (2020) study unlearning through analyzing drift of optimization trajectories built on quadratic loss and determine the weight difference at infinity time limit. Ding et al. (2025) use linear models and weight-space distances to characterize approximate unlearning. Specifically, they use loss as criterion to quantify the difference brought by unlearning process. The recent work (Mavrothalassitis et al., 2025) operates in linear logistic regression, by studying closed solution based on the logistic loss, they describe the unlearning process through the weight difference between original one and the unlearned one. Thudi et al. (2022)analyze unlearning by unrolling the SGD recursion, which again relies on linearization of the gradient dynamics. Also their definition of unlearning error directly use the difference in weight space and connect the MIA attack to this theoretically defined quantity.

## 5.5 Feasibility of computing data coherence

A practical concern raised by reviewers is that the theory's "central metric relies on per-sample Hessians or Gram matrices," which are expensive or infeasible to compute on modern large models. This is a valid point. Exact per-sample Hessians in a deep network can be enormous but it is not a fatal flaw of the approach. There is strong precedent in ML theory where initially intractable quantities inspired new insights and eventually yielded practical approximations. The Fisher Information Matrix (FIM) and the full Hessian of a network are classic examples: early theoretical research treated them as important objects despite their size, and this spurred the development of methods to approximate or constrain them(Martens & Grosse, 2020; Yao et al., 2020). Natural gradient methods and second-order optimizers like K-FAC (Kronecker-Factored Approximate Curvature)(Martens & Grosse, 2020) explicitly approximate the Fisher/Hessian to achieve near-optimal descent directions. In fact, Sharpness-Aware Minimization (SAM)(Foret et al., 2021) (a recent regularizer that improves generalization) was inspired by the idea of penalizing the Hessian's largest eigenvalues (i.e. minimizing sharpness). SAM doesn't compute the full Hessian; it uses a clever first-order approximation (perturbing weights to measure curvature indirectly), yet it stemmed from the principle that the Hessian spectrum matters. Likewise, the Neural Tangent Kernel was originally an N×N Gram matrix over data points which is seemingly impractical beyond toy datasets, but it led to kernel proxies and inspired practical diagnostics. For instance, researchers developed ways to estimate the NTK(Novak et al., 2019) or related Gram matrices for subsets of data to monitor training dynamics, and used the constant-NTK theory to justify why wide networks behave more predictably.

In our case, Hessian alignment/coherence is introduced as a conceptual tool to understand unlearning. While we indeed computed it in a small controlled CNN to validate the theory, this is akin to how many theoretical analyses proceed: first verify on a "toy" setup where the exact metrics can be computed for

clarity, then later work on scaling it up. It's worth noting that many large-scale theoretical studies have found ways to approximate Hessian-based measures. For example, Ghorbani et al. (2019) developed numerical linear algebra techniques to estimate the entire Hessian eigenvalue density for ImageNet-scale networks. They used random matrix sketching and power-iteration methods to produce the Hessian spectrum efficiently, a feat that seemed impossible a few years prior. This underscores that what's "infeasible" with brute force can become feasible with algorithmic ingenuity. The history of deep learning research shows that what starts as "only explanatory" can become actionable. The NTK was once purely theoretical, yet now practitioners talk about "NTK conditioning" and use kernel analogies to choose architectures. Likewise, we anticipate that coherence measures could inspire new diagnostics (perhaps a coherence score computed on a small held-out batch as a proxy) or new training procedures (e.g. encourage decorrelation between forget and retain gradients). In our submission, we acknowledged that directly computing per-sample Hessians for a giant model is impractical today, but we intentionally validated our theory in a setting where we could compute them exactly, thus establishing a clear ground truth. This is a valuable first step. Moving forward, our framework can guide research into scalable approximations: perhaps using low-rank factorization of the Hessian, or computing block-wise coherence (layer-wise, or between specific neural units) as a cheaper metric. Thus, we confidently defend the use of linear stability and local linearization in our analysis: it is a principled approach grounded in prior successes in deep learning theory, and it offers a powerful explanatory framework for understanding when models will – or won't – let go of what they have learned.

## 5.6 MORE DISCUSSION ABOUT MEMORIZATION AND FORGETTING.

Our definition of memorization is grounded in the observation that, to minimize training loss, models often overfit to highly orthogonal components of the data and the directions that are uncorrelated with the main signal. This view aligns with our coherence-based analysis: in our signal-plus-noise experiments, noise components are orthogonal in expectation, and the theory predicts that such directions are easily forgotten once ascent begins.

Similar notions have been explored in recent work. Wen et al. (2023) show that memorized examples correspond to orthogonal activation patterns within the network, which translate into orthogonal Hessian directions, while Yu et al. (2024) study memorization in highly orthogonal subspaces. These results support our geometric interpretation that memorization arises from localized, low-coherence modes.

The type of memorization captured by our coherence framework (fitting orthogonal directions or outlier features) is one of the most fundamental and widely studied forms of memorization in modern deep learning. It is closely connected to optimization stability, generalization behavior, and ultimately forgetting. We agree that verbatim sequence memorization in large language models may involve additional mechanisms; however, the underlying causes of such memorization remain an open research question with no corresponding theoretical formulation and/or studies to the best of our knowledge yet. Because our work focuses on the optimization geometry governing gradient-based learning, extending coherence-based analysis to sequential or long-context memorization in LLMs represents an exciting future direction.

## 5.7 LEMMAS AND PROOFS

**Definition 6** (Full forget Hessian and retain Hessian)**.**

$$H_R = \frac{1}{n_r} \sum_{r \in D_r} H_r, \ \ H_F = \frac{1}{n_r} \sum_{f \in D_f} H_f, \tag{15}$$

**Lemma 5.1.** $l_1$-$l_2$ *norm inequality: For any* $x \in \mathbb{R}, ||x||_2 \leq ||x||_1 \leq \sqrt{d}||x||_2$

**Lemma 5.2.** ***Binomial coefficient:*** *For all* $n, k \in \mathbb{N}$ *such that* $k \leq n$*, the binomial coefficients satisfy that*

$$\binom{n}{k} = \binom{n-1}{k-1} + \binom{n-1}{k}. \tag{16}$$

**Lemma 5.3.** *For any matrix* $M \in \mathbb{R}^{n \times n}$, $||M||_F \leq ||M||_{S_1} \leq \sqrt{n}||M||_F$, *where* $||M||_{S_p}$ *is p norm of the spectrum of M, and the inequality is obtain through* $l_1$-$l_2$ *norm inequality.*

**Lemma 5.4.** *For matrices $M_1...M_k \in \mathbb{R}^{n \times n}$, $\mathrm{Tr}[M_1...M_k] \leq ||M_1...M_k||_{S_1}$ (see Bhatia (2013))*

**Lemma 5.5** (Vershynin (2018))**.** *Consider $n$ random gaussian vectors $x_1...x_n$ sampled i.i.d from $N(0, \sigma^2 I_d)$, there exist a constant $C_1$ such that with probability $1 - \delta$,*

$$\sum_{i=1}^{n} \|x_i\| \leq n\sqrt{2}\sigma \frac{\Gamma((d+1)/2)}{\Gamma(d/2)} + \sqrt{\frac{n\sigma^2}{C_1} \log(\frac{2}{\delta})} \quad \text{for } n, d \text{ large enough.} \tag{17}$$

**Lemma 5.6** (Vershynin (2018))**.** *Consider $n$ random gaussian vectors $x_1...x_n$ sampled i.i.d from $N(0, \sigma^2 I_d)$, there exist a constant $C_1$ such that with probability $1 - \delta$,*

$$\sum_{i=1}^{n} \|x_i\|^2 \leq n\sigma^2 d + \sqrt{\frac{n\sigma^4 d}{C_2} \log(\frac{2}{\delta})} \quad \text{for } n, d \text{ large enough.} \tag{18}$$

### 5.8 PROOF OF LEMMA 3.1

**Lemma 5.7** (Restated)**.** *Consider the unlearning update operator $J_k$ defined in (4). Define a sequence of PSD matrices $\{N_k\}_{k \geq 0}$ by $N_0 = I$ and for $k \geq 1$:*

$$N_k = C_f \sum_{i \in D_f} H_i N_{k-1} H_i + C_r \sum_{i \in D_r} H_i N_{k-1} H_i, \tag{19}$$

*with $C_r, C_f$ as given in Definition 2. Also let $M_k = J^{2k} + N_k$. Then:*

1. *(**Lower bound**) $\mathbb{E}\,\mathrm{Tr}\big(J_0^T \cdots J_{k-1}^T J_{k-1} \cdots J_0\big) \geq \mathrm{Tr}(M_k)$. Moreover, if $\mathrm{Tr}(N_k) \to \infty$ as $k \to \infty$, then $\mathbb{E}\|w_k\|^2 \to \infty$ as well.*

2. *(**Upper bound**) If at each step $J_k$ is spectrally bounded as $(1 - \epsilon)I \succeq J \succeq -(1 - \epsilon)I$ for some $\epsilon \in (0, 1)$ (i.e. all eigenvalues of $J$ lie in $[-(1 - \epsilon), 1 - \epsilon]$), then*

$$\mathbb{E}\,\mathrm{Tr}\big(J_0^T \cdots J_{k-1}^T J_{k-1} \cdots J_0\big) \leq \sum_{r=0}^{k-1} \binom{k}{r}(1 - \epsilon)^{2(k-r)} \mathrm{Tr}(N_r).$$

*If in addition $\mathrm{Tr}(N_r) \leq \epsilon$ for all $r$, then $\mathbb{E}\|w_k\|^2 \to 0$ as $k \to \infty$ (the unlearning update converges in mean square).*

*Proof.* As we are taking the expectation value over the calculation, we can effectively transform the $J_k$ into following with random variables involved:

$$J_k = (I - \eta(1-\alpha)\frac{1}{B} \sum_{i \in D_{r,k}} H_i + \eta\alpha\frac{1}{B} \sum_{i \in D_{f,k}} H_i) = (I - \eta(1-\alpha)\frac{1}{B} \sum_{r \in D_r} x_r H_r + \eta\alpha\frac{1}{B} \sum_{i \in D_f} x_f H_f), \tag{20}$$

where $x_r, x_f$ are the corresponding Bernoulli random variables with probability $P(x_r = 1) = \frac{B}{n_r}$ and $P(x_f = 1) = \frac{B}{n_f}$ and 0 otherwise.

To initiate the first step in characterize the difference between the unlearning and usually learning process, we first calculate the $E[J_1^T J_1]$ as follows:

$$E[J_1^T J_1] = E[(I - \eta(1-\alpha)\frac{1}{B} \sum_{r \in D_r} x_r H_r + \eta\alpha\frac{1}{B} \sum_{i \in D_f} x_f H_f)^T (I - \eta(1-\alpha)\frac{1}{B} \sum_{r \in D_r} x_r H_r + \eta\alpha\frac{1}{B} \sum_{i \in D_f} x_f H_f)]$$

$$= E[(I - \eta(1-\alpha)\frac{1}{B} \sum_{r \in D_r} x_r H_r)^T (I - \eta(1-\alpha)\frac{1}{B} \sum_{r \in D_r} x_r H_r)$$

$$+ 2(I - \eta(1-\alpha)\frac{1}{B} \sum_{r \in D_r} x_r H_r)^T (\eta\alpha\frac{1}{B} \sum_{i \in D_f} x_f H_f) + (\eta\alpha\frac{1}{B} \sum_{i \in D_f} x_f H_f)^T (\eta\alpha\frac{1}{B} \sum_{i \in D_f} x_f H_f)]. \tag{21}$$

Here, we separate the above equation into three part and take the expectation accordingly:

$$E[(I - \eta(1-\alpha)\frac{1}{B}\sum_{r \in D_r} x_r H_r)^T(I - \eta(1-\alpha)\frac{1}{B}\sum_{r \in D_r} x_r H_r)]$$

$$= E[(I - 2\eta(1-\alpha)\frac{1}{B}\sum_{r \in D_r} x_r H_r + \eta^2(1-\alpha)^2\frac{1}{B}^2\sum_{r \in D_r} x_r H_r \sum_{r \in D_r} x_r H_r)],$$

$$= I - 2\eta(1-\alpha)\frac{1}{n_r}\sum_{r \in D_r} H_r + E[\eta^2(1-\alpha)^2(\frac{1}{B})^2\sum_{r \in D_r} x_r H_r \sum_{r \in D_r} x_r H_r],$$

$$= I - 2\eta(1-\alpha)\frac{1}{n_r}\sum_{r \in D_r} H_r + E[\eta^2(1-\alpha)^2(\frac{1}{B})^2\sum_{r' \in D_r}\sum_{r \in D_r} x_{r'} x_r H_{r'} H_r],$$

$$= I - 2\eta(1-\alpha)\frac{1}{n_r}\sum_{r \in D_r} H_r + \eta^2(1-\alpha)^2(\frac{1}{n_r})^2(\sum_{r \in D_r} H_r)^2 + \eta^2(1-\alpha)^2\frac{1}{n_r}(\frac{1}{B} - \frac{1}{n_r})\sum_{r \in D_r} H_r^2,$$

$$= I - 2\eta(1-\alpha)H_R + \eta^2(1-\alpha)^2 H_R^2 + \eta^2(1-\alpha)^2\frac{1}{n_r}(\frac{1}{B} - \frac{1}{n_r})\sum_{r \in D_r} H_r^2,$$

$$= (I - \eta(1-\alpha)H_R)^2 + \eta^2(1-\alpha)^2\frac{1}{n_r}(\frac{1}{B} - \frac{1}{n_r})\sum_{r \in D_r} H_r^2. \tag{22}$$

The random variables $x_r$ are independent to each other but not itself and therefore there exist one additional terms in the final line. Also, Compared to the original sgd there exists additional multiplication of the $(1-\alpha)^2$. Next, we move on to the interaction term:

$$E[2(I - \eta(1-\alpha)\frac{1}{B}\sum_{r \in D_r} x_r H_r)^T(\eta\alpha\frac{1}{B}\sum_{i \in D_f} x_f H_f)] = 2(I - \eta(1-\alpha)H_R)^T(\eta\alpha H_F). \tag{23}$$

We can directly formulate this as above due to the fact that we assume the sampling process of retain set and forget set to be independent. Last, the term arising due to the forget set:

$$E[(\eta\alpha\frac{1}{B}\sum_{f \in D_f} x_f H_f)^T(\eta\alpha\frac{1}{B}\sum_{f \in D_f} x_f H_f)] = \eta^2\alpha^2 H_F^2 + \eta^2\alpha^2\frac{1}{n_f}(\frac{1}{B} - \frac{1}{n_f})\sum_{f \in D_f}^{n_f} H_f^2. \tag{24}$$

We then integrate the three part and reformulate the Jacobian:

$$E[J_1^T J_1] = (I - \eta(1-\alpha)H_R)(I - \eta(1-\alpha)H_R) + 2(I - \eta(1-\alpha)H_R)^T(\eta\alpha H_F) + \eta^2\alpha^2 H_F^2$$

$$+ \eta^2\alpha^2\frac{1}{n_f}(\frac{1}{B} - \frac{1}{n_f})\sum_{f \in D_f}^{n_f} H_f^2 + \eta^2(1-\alpha)^2\frac{1}{n_r}(\frac{1}{B} - \frac{1}{n_r})\sum_{r \in D_r}^{n_r} H_r^2,$$

$$= (I - \eta(1-\alpha)H_R + \eta\alpha H_F)(I - \eta(1-\alpha)H_R + \eta\alpha H_F)$$

$$+ \eta^2\alpha^2\frac{1}{n_f}(\frac{1}{B} - \frac{1}{n_f})\sum_{f \in D_f}^{n_f} H_f^2 + \eta^2(1-\alpha)^2\frac{1}{n_r}(\frac{1}{B} - \frac{1}{n_r})\sum_{r \in D_r}^{n_r} H_r^2,$$

$$= J^2 + \eta^2\alpha^2\frac{1}{n_f}(\frac{1}{B} - \frac{1}{n_f})\sum_{f \in D_f}^{n_f} H_f^2 + \eta^2(1-\alpha)^2\frac{1}{n_r}(\frac{1}{B} - \frac{1}{n_r})\sum_{r \in D_r}^{n_r} H_r^2, \tag{25}$$

where we define $J = I - \eta(1-\alpha)H_R + \eta\alpha H_F$. During the whole work, we will be analyzing on these terms to characterize the behavior of unlearning process.

We use inductive proof for both first and second part of the theory and we begin the proof as following:

**First part:**

**Base case: k = 1**

$$M_1 = J^2 + C_f \sum_{f \in D_f}^{n_f} H_f^2 + C_r \sum_{r \in D_r}^{n_r} H_r^2 = J^2 + N_1 \preceq E[J_1^T J_1], \tag{26}$$

where the left term match the equation 25 and therefore the basis case is set. Now we go further with inductive step.

**Inductive step: k-1**

$$E[J_k^T J_{k-1}^T ... J_1^T J_1 ... J_{k-1}] \succeq E[J_k^T M_{k-1} J_k],$$

$$= E[(I - \eta(1-\alpha)\frac{1}{B}\sum_{i \in D_r} x_r H_r + \eta\alpha\frac{1}{B}\sum_{f \in D_f} x_f H_f)M_{k-1}(I - \eta(1-\alpha)\frac{1}{B}\sum_{i \in D_r} x_r H_r + \eta\alpha\frac{1}{B}\sum_{i \in D_f} x_f H_f)],$$

$$= JM_{k-1}J + C_f \sum_{f \in D_f}^{n_f} H_f M_{k-1} H_f + C_r \sum_{r \in D_r}^{n_r} H_r M_{k-1} H_r,$$

$$= J(J^{2(k-1)} + N_{k-1})J + C_f \sum_{f \in D_f}^{n_f} H_f(J^{2(k-1)} + N_{k-1})H_f + C_r \sum_{r \in D_r}^{n_r} H_r(J^{2(k-1)} + N_{k-1})H_r,$$

$$= J^{2k} + C_f \sum_{f \in D_f}^{n_f} H_f N_{k-1} H_f + C_r \sum_{r \in D_r}^{n_r} H_r N_{k-1} H_r + JN_{k-1}J + C_f \sum_{f \in D_f}^{n_f} H_f J^{2(k-1)} H_f + C_r \sum_{r \in D_r}^{n_r} H_r J^{2(k-1)} H_r,$$

$$= M_k + JN_{k-1}J + C_f \sum_{f \in D_f}^{n_f} H_f J^{2(k-1)} H_f + C_r \sum_{r \in D_r}^{n_r} H_r J^{2(k-1)} H_r,$$

$$\succeq M_k. \tag{27}$$

The last equality is due to the later three terms are both PSD by assumption as they are symmetric in terms of left and right half of whole multiplication. As we can lower bound through $M_k$, diverging of $N_k$ will lead to $M_k$ and cause the whole product to diverge.

**Second part:**

**Base step: k=1.**

$$E[J_1^T J_1] = J^2 + N_1 \preceq (1-\epsilon)^2 I + N_1 = \sum_{r=0}^{1} \binom{1}{r}(1-\epsilon)^{2(1-r)} N_r. \tag{28}$$

The $J^2$ is bounded by $(1-\epsilon)^2 I$ due to our assumption. Now, we start with the inductive step

**Inductive step: k-1.**

$$E[J_k^T J_{k-1}^T ... J_1^T J_1 ... J_{k-1} J_{k-1}] \preceq E[J_k^T (\sum_{r=0}^{k-1} \binom{k-1}{r} (1-\epsilon)^{2(k-1-r)} N_r) J_k],$$

$$= J(\sum_{r=0}^{k-1} \binom{k-1}{r} (1-\epsilon)^{2(k-1-r)} N_r) J$$

$$+ C_f \sum_{i \in D_f}^{n_f} H_i (\sum_{r=0}^{k-1} \binom{k-1}{r} (1-\epsilon)^{2(k-1-r)} N_r) H_i + C_r \sum_{i \in D_r}^{n_r} H_i (\sum_{r=0}^{k-1} \binom{k-1}{r} (1-\epsilon)^{2(k-1-r)} N_r) H_i,$$

$$= J(\sum_{r=0}^{k-1} \binom{k-1}{r} (1-\epsilon)^{2(k-1-r)} N_r) J + \sum_{r=0}^{k-1} \binom{k-1}{r} (1-\epsilon)^{2(k-1-r)} (C_f \sum_{i \in D_f}^{n_f} H_i N_r H_i + C_r \sum_{i \in D_r}^{n_r} H_i N_r H_i),$$

$$\preceq \sum_{r=0}^{k-1} \binom{k-1}{r} (1-\epsilon)^{2(k-r)} N_r + \sum_{r=0}^{k-1} \binom{k-1}{r} (1-\epsilon)^{2(k-1-r)} N_{r+1},$$

$$= \sum_{r=0}^{k-1} \binom{k-1}{r} (1-\epsilon)^{2(k-r)} N_r + \sum_{r=0}^{k-1} \binom{k-1}{r} (1-\epsilon)^{2(k-1-r)} N_{r+1},$$

$$= (1-\epsilon)^2 N_o + \sum_{r=1}^{k-1} (\binom{k-1}{r} + \binom{k-1}{r-1}) N_r + N_k,$$

$$= (1-\epsilon)^2 N_o + \sum_{r=1}^{k-1} \binom{k}{r} N_r + N_k,$$

$$= \sum_{r=0}^{k} \binom{k}{r} (1-\epsilon)^{2(k-r)} N_r.$$

$$(29)$$

$$\square$$

The first and second inequality is due to the assumption in induction on previous step and we merge the coefficient in the last step through lemma 5.2. Finally, if we further have that $\mathrm{Tr}[N_r] \leq \epsilon \ \forall r$, then

$$E[\mathrm{Tr}[J_k^T J_{k-1}^T ... J_1^T J_1 ... J_{k-1} J_{k-1}]],$$

$$= \sum_{r=0}^{k} \binom{k}{r} (1-\epsilon)^{2k-r} \mathrm{Tr}[N_r],$$

$$\leq \sum_{r=0}^{k} \binom{k}{r} (1-\epsilon)^{2(k-r)} \epsilon^r,$$

$$\leq ((1-\epsilon)^2 + \epsilon)^k \leq (1-\epsilon)^k,$$

$$(30)$$

which will converge to zero when $k \to \infty$.

## 5.9 Proof of theorem 3.2

**Theorem 5.8** (Restated). *Under the setup of Lemma 3.1, the unlearning process will diverge if the mix-Hessian eigenvalue exceeds a threshold determined by the coherence. In particular, if*

$$\lambda_{\max}(D) \geq \frac{\sqrt{2}\,\sigma}{\eta\Big((1-\alpha)\,n_f\,\sqrt{\frac{n_r}{B}-1} + \alpha\,n_r\,\sqrt{\frac{n_f}{B}-1}\Big)}\,, \tag{31}$$

*then* $\lim_{k\to\infty} \mathbb{E}\|w_k\|^2 = \infty$. *Equivalently, condition (8) guarantees the unlearning algorithm will escape the original minima (diverge) due to the stochastic dynamics.*

*Proof.* To simplify the notation, we use the following

$$\begin{aligned}
L_k &\in \{r,f\}^k \quad \text{(a string of length } k \text{ over the alphabet } \{r,f\}), \\
L_k[i] &\mapsto \quad \text{the } i\text{-th symbol of } L_k, \quad 1 \leq i \leq k.
\end{aligned} \tag{32}$$

We know that based on part one lemma 3.1, we can lower bound the $N_k$ to lower bound the $E[\text{Tr}[J_k^T J_{k-1}^T ... J_1^T J_1 ... J_{k-1} J_k]]$. First, We can write the overall sum as follows:

$$\begin{aligned}
\text{Tr}[N_k] &= \sum_{L_k \in \{r,f\}^k} C_r^{\sum_{i=1}^k \mathbf{1}\{L_k[i]=r\}} C_f^{\sum_{i=1}^k \mathbf{1}\{L_k[i]=f\}} \Big( \sum_{a_k \in D_{L_k[k]}} \sum_{a_{k-1} \in D_{L_k[k-1]}} \cdots \sum_{a_1 \in D_{L_k[k]}} \Big) \text{Tr}[H_{a_k}...H_{a_1} H_{a_1}...H_{a_k}], \\
&= \sum_{L_k \in \{r,f\}^k} C_r^{\sum_{i=1}^k \mathbf{1}\{L_k[i]=r\}} C_f^{\sum_{i=1}^k \mathbf{1}\{L_k[i]=f\}} \Big( \sum_{a_k \in D_{L_k[k]}} \sum_{a_{k-1} \in D_{L_k[k-1]}} \cdots \sum_{a_1 \in D_{L_k[k]}} \Big) \|H_{a_k}...H_{a_1}\|_F^2, \\
&\geq \frac{1}{d} \sum_{L_k \in \{r,f\}^k} C_r^{\sum_{i=1}^k \mathbf{1}\{L_k[i]=r\}} C_f^{\sum_{i=1}^k \mathbf{1}\{L_k[i]=f\}} \Big( \sum_{a_k \in D_{L_k[k]}} \sum_{a_{k-1} \in D_{L_k[k-1]}} \cdots \sum_{a_1 \in D_{L_k[k]}} \Big) \|H_{a_k}...H_{a_1}\|_{S_1}^2, \\
&\geq \frac{1}{d} \sum_{L_k \in \{r,f\}^k} C_r^{\sum_{i=1}^k \mathbf{1}\{L_k[i]=r\}} C_f^{\sum_{i=1}^k \mathbf{1}\{L_k[i]=f\}} \Big( \sum_{a_k \in D_{L_k[k]}} \sum_{a_{k-1} \in D_{L_k[k-1]}} \cdots \sum_{a_1 \in D_{L_k[k]}} \Big) \text{Tr}[H_{a_k}...H_{a_1}]^2, \\
&\geq \frac{1}{n_r n_f} \frac{1}{d} \sum_{L_k \in \{r,f\}^k} C_r^{\sum_{i=1}^k \mathbf{1}\{L_k[i]=r\}} C_f^{\sum_{i=1}^k \mathbf{1}\{L_k[i]=f\}} \Big( \sum_{a_r \in D_r} \sum_{a_f \in D_f} \Big) \text{Tr}[H_{a_{L_k[k]}}...H_{a_{L_k[1]}}]^2, \\
&= \frac{1}{n_r n_f} \frac{1}{d} \Big( \sum_{a_r \in D_r} \sum_{a_f \in D_f} \Big) \sum_{L_k \in \{r,f\}^k} C_r^{\sum_{i=1}^k \mathbf{1}\{L_k[i]=r\}} C_f^{\sum_{i=1}^k \mathbf{1}\{L_k[i]=f\}} \text{Tr}[H_{a_{L_k[k]}}...H_{a_{L_k[1]}}]^2, \\
&\geq \frac{1}{n_r n_f} \frac{1}{d} \Big( \sum_{a_r \in D_r} \sum_{a_f \in D_f} \Big) \frac{1}{2^k} \Big( \sum_{L_k \in \{r,f\}^k} \text{Tr}[C_r^{\frac{\sum_{i=1}^k \mathbf{1}\{L_k[i]=r\}}{2}} C_f^{\frac{\sum_{i=1}^k \mathbf{1}\{L_k[i]=f\}}{2}} H_{a_{L_k[k]}}...H_{a_{L_k[1]}}] \Big)^2, \\
&= \frac{1}{n_r n_f} \frac{1}{d} \Big( \sum_{a_r \in D_r} \sum_{a_f \in D_f} \Big) \frac{1}{2^k} \Big( \text{Tr}[\sum_{L_k \in \{r,f\}^k} C_r^{\frac{\sum_{i=1}^k \mathbf{1}\{L_k[i]=r\}}{2}} C_f^{\frac{\sum_{i=1}^k \mathbf{1}\{L_k[i]=f\}}{2}} H_{a_{L_k[k]}}...H_{a_{L_k[1]}}] \Big)^2, \\
&= \frac{1}{n_r n_f} \frac{1}{d} \Big( \sum_{a_r \in D_r} \sum_{a_f \in D_f} \Big) \frac{1}{2^k} (C_r^{\frac{1}{2}} + C_f^{\frac{1}{2}})^{2k} \Big( \text{Tr}[(\frac{C_r^{\frac{1}{2}}}{C_r^{\frac{1}{2}} + C_f^{\frac{1}{2}}} H_{a_r} + \frac{C_f^{\frac{1}{2}}}{C_r^{\frac{1}{2}} + C_f^{\frac{1}{2}}} H_{a_f})^k] \Big)^2, \\
&= \Big(\frac{1}{n_r n_f}\Big)^2 \frac{1}{d} \frac{1}{2^k} (C_r^{\frac{1}{2}} + C_f^{\frac{1}{2}})^{2k} \Big( \sum_{a_r \in D_r} \sum_{a_f \in D_f} \text{Tr}[(\frac{C_r^{\frac{1}{2}}}{C_r^{\frac{1}{2}} + C_f^{\frac{1}{2}}} H_{a_r} + \frac{C_f^{\frac{1}{2}}}{C_r^{\frac{1}{2}} + C_f^{\frac{1}{2}}} H_{a_f})^k] \Big)^2.
\end{aligned} \tag{33}$$

For the first and second inequality, we use lemma 5.3 and 5.4. For the third inequality, we reduce the summation to $\sum_{a_r \in D_r} \sum_{a_f \in D_f}$. As there are terms without $D_r$ or $D_f$ involved, we divided the whole equation by $n_f n_r$ to ensure inequality. For the forth inequality, we use the lemma 5.1.

Before we try to connect the relationship between the quantity to the above, we first reindex the following:

$$\frac{1}{n_r n_f} \sum_{a_r \in D_r, a_f \in D_f} \frac{C_r^{\frac{1}{2}}}{C_r^{\frac{1}{2}} + C_f^{\frac{1}{2}}} H_{a_r} + \frac{C_f^{\frac{1}{2}}}{C_r^{\frac{1}{2}} + C_f^{\frac{1}{2}}} H_{a_f} = \frac{1}{n_r n_f} \sum_{rf} D_{rf} = D, \qquad (34)$$

where $D_{rf} = \frac{C_r^{\frac{1}{2}}}{C_r^{\frac{1}{2}} + C_f^{\frac{1}{2}}} H_r + \frac{C_f^{\frac{1}{2}}}{C_r^{\frac{1}{2}} + C_f^{\frac{1}{2}}} H_f$ and the subscript indicates that summing over corresponding subset (retain and forget set). Now, we proceed to relate different quantities

$$\mathrm{Tr}[(\frac{1}{n_r n_f} \sum_{a_r \in D_r, a_f \in D_f} \frac{C_r^{\frac{1}{2}}}{C_r^{\frac{1}{2}} + C_f^{\frac{1}{2}}} H_{a_r} + \frac{C_f^{\frac{1}{2}}}{C_r^{\frac{1}{2}} + C_f^{\frac{1}{2}}} H_{a_f})^k] = \mathrm{Tr}[(\frac{1}{n_r n_f})^k (\sum_{rf} D_{rf})^k],$$

$$= \mathrm{Tr}[(\frac{1}{n_r n_f})^k \sum_{rf_1} \sum_{rf_2} ... \sum_{rf_k} D_{rf_1} D_{rf_2} ... D_{rf_{k-1}} D_{rf_k}],$$

$$\leq d(\frac{1}{n_r n_f})^k \sum_{rf_1} \sum_{rf_2} ... \sum_{rf_k} \|D_{rf_k}^{\frac{1}{2}} D_{rf_1}^{\frac{1}{2}}\|_F \|D_{rf_1}^{\frac{1}{2}} D_{rf_2}^{\frac{1}{2}}\|_F ... \|D_{rf_{k-1}}^{\frac{1}{2}} D_{rf_k}^{\frac{1}{2}}\|_F,$$

$$= d(\frac{1}{n_r n_f})^k \sum_{rf_1} \sum_{rf_2} ... \sum_{rf_k} S_{rf_k, rf_1} S_{rf_1, rf_2} ... S_{rf_{k-1}, rf_k}, \qquad (35)$$

$$= d(\frac{1}{n_r n_f})^k \mathrm{Tr}(S^k),$$

$$\leq d^2 (\frac{1}{n_r n_f})^k \lambda_1(S)^k.$$

Therefore, we say that

$$\mathrm{Tr}[D^k] \leq d^2 (\frac{1}{n_r n_f})^k \lambda_1(S)^k, \qquad (36)$$

and we can have that

$$\frac{(n_r n_f)^k \mathrm{Tr}[D^k]}{d^2 \sigma^k} \leq \frac{(n_r n_f)^k \mathrm{Tr}[D^k]}{d^2 \lambda_1(S)^k} \max_{i \in D_r, j \in D_f} \lambda_1 \Big(\frac{C_r^{\frac{1}{2}}}{C_r^{\frac{1}{2}} + C_f^{\frac{1}{2}}} H_i + \frac{C_f^{\frac{1}{2}}}{C_r^{\frac{1}{2}} + C_f^{\frac{1}{2}}} H_j\Big)^k,$$

$$\leq \sum_{a_r \in D_r} \sum_{a_f \in D_f} \mathrm{Tr}[(\frac{C_r^{\frac{1}{2}}}{C_r^{\frac{1}{2}} + C_f^{\frac{1}{2}}} H_{a_r} + \frac{C_f^{\frac{1}{2}}}{C_r^{\frac{1}{2}} + C_f^{\frac{1}{2}}} H_{a_f})^k]. \qquad (37)$$

Therefore, we can conclude that

$$\mathrm{Tr}[N_k] \geq \frac{1}{d} \frac{1}{n_f n_r} \frac{1}{2^k} (C_r^{\frac{1}{2}} + C_f^{\frac{1}{2}})^{2k} (\frac{(n_r n_f)^k \mathrm{Tr}[D^k]}{d^2 \sigma^k})^2,$$

$$\geq \frac{1}{d^5} \frac{1}{n_f n_r} \frac{1}{2^k} (C_r^{\frac{1}{2}} + C_f^{\frac{1}{2}})^{2k} \frac{(n_r n_f)^{2k} \lambda_1(D)^{2k}}{\sigma^{2k}}, \qquad (38)$$

$$\geq \frac{1}{d^5} \frac{1}{n_f n_r} \frac{1}{2^k} (C_r^{\frac{1}{2}} + C_f^{\frac{1}{2}})^{2k} \frac{(n_r n_f)^{2k} \lambda_1(D)^{2k}}{\sigma^{2k}}.$$

Lastly, we can see that whether the trace diverge or not depend on those term with power of $k$. Therefore, by rearranging and plug in the definition of the coefficient into those terms, we can have that

$$\lambda_1(D) \geq \frac{\sqrt{2}\sigma}{\eta} \Big((1-\alpha) n_f (\frac{n_r}{B} - 1)^{\frac{1}{2}} + \alpha n_r (\frac{n_f}{B} - 1)^{\frac{1}{2}}\Big)^{-1}, \qquad (39)$$

which is the condition for diverging behavior $\qquad \square$

### 5.10 PROOF OF THEOREM 3.3

**Theorem 5.9** ((Restate) Matching lower bound.). *Suppose* $\lambda_{\max}(D)$ *and* $\sigma$ *satisfy*

$$\lambda_{\max}(D) \ \leq \ \frac{2\,\sigma}{\eta\,C_r'\left(\sigma + n_f\left(\frac{n_r}{B} - 1\right)\right)}\,, \tag{40}$$

*where* $C_r' = \sqrt{C_r}/(\sqrt{C_r} + \sqrt{C_f})$ *(with* $C_r, C_f$ *from Definition 2). Then there exists a choice of PSD Hessians* $\{H_i\}$ *for the retain and forget sets such that the unlearning update converges (i.e.* $\lim_{k\to\infty} \mathbb{E}\|w_k\|^2 = 0$) *under those Hessians.*

*Proof.* We prove by construction in the following manner. We construct the retain set by setting $H_i = m \cdot e_1 e_1^T \ \forall\, i \ \in [\frac{\sigma}{n_f}]$. ($m = (C_r')^{-1}\frac{\lambda_1(D)n_r}{\frac{\sigma}{n_f}}$ and $C_r' = \frac{C_r^{\frac{1}{2}}}{C_r^{\frac{1}{2}} + C_f^{\frac{1}{2}}}$ and the definition of $C_r$ and $C_f$ are mentioned in definition 2.) Otherwise, we set the Hessian to be zero matrix. For the forget set, we set all matrix to be zero matrix.

We first verify that the eigenvalue of mix-Hessian is indeed the assigned value $\lambda_1(D)$.

$$D = \frac{1}{n_r n_f} \sum_{rf} C_r' H_r + C_f' H_f = \frac{1}{n_r n_f} \sum_{rf} C_r' (C_r')^{-1} \frac{\lambda_1(D)n_r}{\frac{\sigma}{n_f}} e_1 e_1^T = \lambda_1(D) e_1 e_1^T, \tag{41}$$

and we have that the construction indeed have the corresponding mix-Hessian eigenvalue.

We know verify that the coherence measure is of the assigned value $\sigma$. We first note that the element of the coherence matrix is:

$$S_{rf,r'f'} = \sqrt{\mathrm{Tr}[(C_r' H_r + C_f' H_f)(C_r' H_{r'} + C_f' H_{f'})]} = C_r' m = \frac{\lambda_1(D)n_r}{\frac{\sigma}{n_f}}, \ \ \forall r, r' \in [\frac{\sigma}{n_f}]. \tag{42}$$

else it is zero. We know that there is $n_f \cdot \frac{\sigma}{n_f} = \sigma$ nonzero elements for each row and column. We note that we will also need to divide the coherence matrix by $\max_{rf} D_{rf} = \max_{rf} C_r' H_r + C_f' H_f = \frac{\lambda_1(D)n_r}{\frac{\sigma}{n_f}}$. Finally, each element is 1 after this division, and we can get the eigenvalue of the matrix to be $\sigma$ and verify that the construction is valid.

Now, we note that in our construction, we have each step $J_i$ to commute to each other since every matrix involved is diagonal, so we can focus on one step to calculate the condition that lead to diverging or converging and since we only intentionally set our matrix to be one dimensional, we can study behavior on only one axis $e_1$ by plugging in the above as follows:

$$e_1 E[J_1 J_1] e_1 = e_1 [I - 2\eta(1-\alpha)H_R + \eta^2(1-\alpha)^2 H_R^2 + \eta^2(1-\alpha)^2 \sum_r H_r^2] e_1,$$

$$= 1 - 2\eta(1-\alpha)(C_r')^{-1}\lambda_1(D) + \eta^2(1-\alpha)^2(C_r')^{-2}\lambda_1(D)^2 + \frac{(C_r')^{-2}}{\sigma}\lambda_1(D)^2\eta^2(1-\alpha)^2 n_f(\frac{n_r}{B} - 1). \tag{43}$$

As we want to study the converging behavior, we want the above to be smaller than 1 to have repetitive multiplication lead to converging.

$$1 - 2\eta(1-\alpha)(C_r')^{-1}\lambda_1(D) + \eta^2(1-\alpha)^2(C_r')^{-2}\lambda_1(D)^2 + \frac{(C_r')^{-2}}{\sigma}\lambda_1(D)^2\eta^2(1-\alpha)^2 n_f(\frac{n_r}{B} - 1) \leq 1,$$

$$\implies 2 \geq \eta(1-\alpha)(C_r')^{-1}\lambda_1(D)(1 + \frac{n_f}{\sigma}(\frac{n_r}{B} - 1)),$$

$$\implies 2 \geq \frac{\eta}{\sigma}(1-\alpha)(C_r')^{-1}\lambda_1(D)(\sigma + n_f(\frac{n_r}{B} - 1)),$$

$$\implies \lambda_1(D) \leq \frac{2\sigma}{\eta}C_r'\Big((1-\alpha)(\sigma + n_f(\frac{n_r}{B} - 1))^{-1}\Big). \tag{44}$$

$\square$

### 5.11 PROOF OF THEOREM 5.11

**Theorem 5.10** (Restate). *Under the data model of Definition 5 and the two-layer ReLU CNN defined above, suppose the network is trained to near-zero training loss. Then with probability at least $1 - 8\delta$ (over the random draw of the dataset), the largest eigenvalue of the coherence matrix $S$ for the retain/forget split satisfies*

$$\lambda_{\max}(S) \ \leq \ \mathcal{O}\Big(n_r\, n_f\, d\sigma^2 \Big[(\sqrt{C_r'} + \sqrt{C_f'})^2\, (\mathrm{SNR})^2 + (C_r' + C_f')\Big]\Big), \tag{45}$$

$$\max_{rf} \lambda_{\max}(D_{rf}) \leq \mathcal{O}((C_r' + C_f')(d\sigma^2(\mathrm{SNR})^2 + 1)), \tag{46}$$

*where $C_r'$ and $C_f'$ are the normalized retain/forget weight fractions as defined in Theorem 3.3. Consider division of two quantities and we can find that for small SNR limit and large SNR limit:*

$$\lim_{SNR \to 0} \frac{\lambda_{\max}(S)^{upper}}{\max_{rf} D_{rf}^{upper}} = \mathcal{O}(n_r n_f) \ , \lim_{SNR \to \infty} \frac{\lambda_{\max}(S)^{upper}}{\max_{rf} D_{rf}^{upper}} = \mathcal{O}(n_r n_f(1 + \frac{2\sqrt{C_r' C_f'}}{C_r' + C_f'})). \tag{47}$$

*Proof.* We first calculate the gradient of one sample respective to one of the $w_{j,r}$.

$$\frac{\partial \ell(y_i \cdot f(W, x_i))}{\partial w_{j,r}} = \ell_i' \cdot \frac{j}{m} \cdot (\mathbf{1}_{\{\langle w_{j,r}, y_i \cdot \boldsymbol{\mu}\rangle > 0\}} \cdot \boldsymbol{\mu} + \mathbf{1}_{\{\langle w_{j,r}, \xi_i\rangle > 0\}} \cdot y_i \cdot \xi_i). \tag{48}$$

There are several index in the above equation (i.e., $j$ and $r$) which we use to take derivative with respect to a specific feature weight vector. We will continue to use this notation for future calculation. Now, we move to calculate the second derivative with respect to two different feature of weights for data $i$ as follows:

$$\frac{\partial^2 \ell(y_i \cdot f(W, x_i))}{\partial w_{j,r} \partial w_{j',r'}} =$$

$$\ell_i'' \cdot \frac{jj'}{m^2} \cdot (\mathbf{1}_{\{\langle w_{j,r}, y_i \cdot \boldsymbol{\mu}\rangle > 0\}} \cdot \boldsymbol{\mu} + \mathbf{1}_{\{\langle w_{j,r}, \xi_i\rangle > 0\}} \cdot y_i \cdot \xi_i)(\mathbf{1}_{\{\langle w_{j',r'}, y_i \cdot \boldsymbol{\mu}\rangle > 0\}} \cdot \boldsymbol{\mu} + \mathbf{1}_{\{\langle w_{j',r'}, \xi_i\rangle > 0\}} \cdot y_i \cdot \xi_i)^T. \tag{49}$$

The above is one block of the Hessian. In the following, we will simplify the notation for indicator function (derivative of ReLU) to $\mathbf{1}_{j',r',y_i \cdot \boldsymbol{\mu}}$ and $\mathbf{1}_{j',r',\boldsymbol{\xi}}$ to ease the heavy notation. To calculate the coherence matrix, we need to calculate trace of Hessian product for different sample,

$$\text{Tr}[H_i H_k] = \sum_{j,j',r,r'} \text{Tr}[\frac{\partial^2 \ell(y_i \cdot f(W,x_i))}{\partial w_{j,r} \partial w_{j',r'}} \frac{\partial^2 \ell(y_k \cdot f(W,x_k))}{\partial w_{j',r'} \partial w_{j,r}}],$$

$$= \frac{\ell_i'' \ell_k''}{m^4} \sum_{j,j',r,r'} (\mathbf{1}_{j,r,y_k \cdot \boldsymbol{\mu}} \cdot \boldsymbol{\mu} + \mathbf{1}_{j,r,\xi_k} \cdot y_k \cdot \xi_k)^T (\mathbf{1}_{j,r,y_i \cdot \boldsymbol{\mu}} \cdot \boldsymbol{\mu} + \mathbf{1}_{j,r,\xi_i} \cdot y_i \cdot \xi_i)$$

$$(\mathbf{1}_{j',r',y_i \cdot \boldsymbol{\mu}} \cdot \boldsymbol{\mu} + \mathbf{1}_{j',r',\xi_i} \cdot y_i \cdot \xi_i)^T (\mathbf{1}_{j',r',y_k \cdot \boldsymbol{\mu}} \cdot \boldsymbol{\mu} + \mathbf{1}_{j',r',\xi_k} \cdot y_k \cdot \xi_k),$$

$$= \frac{\ell_i'' \ell_k''}{m^4} \Big( (\sum_{j,r} \mathbf{1}_{j,r,y_k \cdot \boldsymbol{\mu}} \mathbf{1}_{j,r,y_i \cdot \boldsymbol{\mu}}) \|\boldsymbol{\mu}\|^2) + (\sum_{j,r} \mathbf{1}_{j,r,y_k \cdot \boldsymbol{\mu}} \mathbf{1}_{j,r,\xi_k}) \boldsymbol{\mu}^T \xi_k +$$

$$(\sum_{j,r} \mathbf{1}_{j,r,y_k \cdot \boldsymbol{\mu}} \mathbf{1}_{j,r,\xi_i}) \boldsymbol{\mu}^T \xi_i + (\sum_{j,r} \mathbf{1}_{j,r,\xi_k} \mathbf{1}_{j,r,\xi_i}) \xi_k^T \xi_i \Big), \tag{50}$$

$$\Big( (\sum_{j,r} \mathbf{1}_{j',r',y_k \cdot \boldsymbol{\mu}} \mathbf{1}_{j',r',y_i \cdot \boldsymbol{\mu}}) \|\boldsymbol{\mu}\|^2) + (\sum_{j',r'} \mathbf{1}_{j',r',y_k \cdot \boldsymbol{\mu}} \mathbf{1}_{j',r',\xi_k}) \boldsymbol{\mu}^T \xi_k +$$

$$(\sum_{j',r'} \mathbf{1}_{j',r',y_k \cdot \boldsymbol{\mu}} \mathbf{1}_{j',r',\xi_i}) \boldsymbol{\mu}^T \xi_i + (\sum_{j',r'} \mathbf{1}_{j',r',\xi_k} \mathbf{1}_{j',r',\xi_i}) \xi_k^T \xi_i \Big),$$

$$\leq 4 \frac{\ell_i'' \ell_k''}{m^2} (\|\boldsymbol{\mu}\|^2 + |\boldsymbol{\mu}^T \xi_k| + |\boldsymbol{\mu}^T \xi_i| + |\xi_i^T \xi_k|)^2,$$

$$\leq \frac{4}{m^2} (\|\boldsymbol{\mu}\|^2 + |\boldsymbol{\mu}^T \xi_k| + |\boldsymbol{\mu}^T \xi_i| + |\xi_i^T \xi_k|)^2.$$

We now analyze each term in the coherence matrix.

$$S_{r_1 f_{1'}, r_2 f_{2'}} = \sqrt{\text{Tr}((C_r' H_{r_1} + C_f' H_{f_{1'}})(C_r' H_{r_2} + C_f' H_{f_{2'}}))},$$

$$= \sqrt{\text{Tr}[C_r'^2 H_{r_1} H_{r_2}] + \text{Tr}[C_r' C_f' H_{r_1} H_{f_{2'}}] + \text{Tr}[C_r' C_f' H_{f_{1'}} H_{r_2}] + \text{Tr}[C_f'^2 H_{f_{1'}} H_{r_{2'}}]},$$

$$\leq \sqrt{\text{Tr}[C_r'^2 H_{r_1} H_{r_2}]} + \sqrt{\text{Tr}[C_r' C_f' H_{r_1} H_{f_{2'}}]} + \sqrt{\text{Tr}[C_r' C_f' H_{f_{1'}} H_{r_2}]} + \sqrt{\text{Tr}[C_f'^2 H_{f_{1'}} H_{r_{2'}}]}, \tag{51}$$

where the $C_r'$ and $C_f'$ are respectively the normalized coefficient mentioned in the previous section.

As our goat is to estimate the largest eigenvalue of the coherence matrix and its relation between different variables in the design. To estimate the largest eigenvalue, we incur $\epsilon$-net that is used random matrix theory

$$\lambda_1 = \sup_{\|x\|=1} \langle x, Sx \rangle. \tag{52}$$

For one vector $x$, we can write the expression as summation:

$$\langle x, Sx \rangle = \sum_{r_1 f_{1'}, r_2 f_{2'}} S_{r_1 f_{1'}, r_2 f_{2'}} x_{r_1 f_{1'}} x_{r_2 f_{2'}},$$

$$\leq \sum_{r_1 f_{1'}, r_2 f_{2'}} (\sqrt{\text{Tr}[C_r'^2 H_{r_1} H_{r_2}]} + \sqrt{\text{Tr}[C_r' C_f' H_{r_1} H_{f_{2'}}]} + \sqrt{\text{Tr}[C_r' C_f' H_{f_{1'}} H_{r_2}]} + \sqrt{(\text{Tr}[C_f'^2 H_{f_{1'}} H_{r_{2'}}])} x_{r_1 f_{1'}} x_{r_2 f_{2'}}. \tag{53}$$

We can estimate the above through the random matrix theory and upper bound the largest eigenvalue through the elementwise calculation that we set up and use the tail bound for each random variable to provide relationship between controlled variable and the resulting largest eigenvalue. We first separate the discussion into several cases. First case, when we have four different samples $r_1, r_2, f_1', f_2'$, we can have that

$$(\sqrt{\text{Tr}[C_r'^2 H_{r_1} H_{r_2}]} + \sqrt{\text{Tr}[C_r' C_f' H_{r_1} H_{f_{2'}}]} + \sqrt{\text{Tr}[C_r' C_f' H_{f_{1'}} H_{r_2}]} + \sqrt{\text{Tr}[C_f'^2 H_{f_{1'}} H_{r_{2'}}]}) x_{r_1 f_{1'}}, x_{r_2 f_{2'}},$$

$$\leq (C_r' \frac{2}{m}(\|\boldsymbol{\mu}\|^2 + |\boldsymbol{\mu}^T \xi_{r1}| + |\boldsymbol{\mu}^T \xi_{r2}| + |\xi_{r1}^T \xi_{r2}|) + \sqrt{C_r' C_f'} \frac{2}{m}(\|\boldsymbol{\mu}\|^2 + |\boldsymbol{\mu}^T \xi_{r1}| + |\boldsymbol{\mu}^T \xi_{f2'}| + |\xi_{r1}^T \xi_{f2'}|),$$

$$+ \sqrt{C_r' C_f'} \frac{2}{m}(\|\boldsymbol{\mu}\|^2 + |\boldsymbol{\mu}^T \xi_{r2}| + |\boldsymbol{\mu}^T \xi_{f1'}| + |\xi_{f1'}^T \xi_{r2}|) + C_f' \frac{2}{m}(\|\boldsymbol{\mu}\|^2 + |\boldsymbol{\mu}^T \xi_{f2'}| + |\boldsymbol{\mu}^T \xi_{f1'}| + |\xi_{f1'}^T \xi_{f2'}|)) x_{r_1 f_{1'}}, x_{r_2 f_{2'}},$$

$$\leq (\sqrt{C_r'}\|\boldsymbol{\mu}\| + \sqrt{C_f'}\|\boldsymbol{\mu}\| + \sqrt{C_r'}\|\xi_{r1}\| + \sqrt{C_f'}\|\xi_{f1'}\|)(\sqrt{C_r'}\|\boldsymbol{\mu}\| + \sqrt{C_f'}\|\boldsymbol{\mu}\| + \sqrt{C_r'}\|\xi_{r2}\| + \sqrt{C_f'}\|\xi_{f2'}\|) x_{r_1 f_{1'}}, x_{r_2 f_{2'}}.$$

$$(54)$$

Our aiming in the above is to establish relationship between different variables used in the CNN network. In the above, we can see that we can upper bound the eigenvalue by the cross product of the vector $v_{rf} = \sqrt{C_r'}\|\boldsymbol{\mu}\| + \sqrt{C_f'}\|\boldsymbol{\mu}\| + \sqrt{C_r'}\|\xi_{r1}\| + \sqrt{C_f'}\|\xi_{f1'}\|$ since the coherence matrix is upper bound elementwise by the vector. (i.e., $\lambda_1(S) \leq \lambda_1(vv^T) = \|v^T v\|^2$) and this turns the estimation of the eigenvalue into estimation of the magnitude of the vector.

Now, we analyze the $v^T v$,

$$v^T v = \sum_{rf} (\sqrt{C_r'}\|\boldsymbol{\mu}\| + \sqrt{C_f'}\|\boldsymbol{\mu}\| + \sqrt{C_r'}\|\xi_{r1}\| + \sqrt{C_f'}\|\xi_{f1'}\|)(\sqrt{C_r'}\|\boldsymbol{\mu}\| + \sqrt{C_f'}\|\boldsymbol{\mu}\| + \sqrt{C_r'}\|\xi_{r1}\| + \sqrt{C_f'}\|\xi_{f1'}\|),$$

$$= \sum_{rf} (\sqrt{C_r'}\|\boldsymbol{\mu}\| + \sqrt{C_f'}\|\boldsymbol{\mu}\|)^2 + 2(\sqrt{C_r'}\|\boldsymbol{\mu}\| + \sqrt{C_f'}\|\boldsymbol{\mu}\|)(\sqrt{C_r'}\|\xi_{r1}\| + \sqrt{C_f'}\|\xi_{f1'}\|) + (\sqrt{C_r'}\|\xi_{r1}\| + \sqrt{C_f'}\|\xi_{f1'}\|)^2,$$

$$= n_r n_f (\sqrt{C_r'}\|\boldsymbol{\mu}\| + \sqrt{C_f'}\|\boldsymbol{\mu}\|)^2 + 2(\sqrt{C_r'}\|\boldsymbol{\mu}\| + \sqrt{C_f'}\|\boldsymbol{\mu}\|) \sum_{rf} (\sqrt{C_r'}\|\xi_{r1}\| + \sqrt{C_f'}\|\xi_{f1'}\|)+$$

$$\sum_{rf} (C_r'\|\xi_{r1}\|^2 + C_f'\|\xi_{f1'}\|^2 + \sqrt{C_r' C_f'}'\|\xi_{r1}\|\|\xi_{f1'}\|).$$

$$(55)$$

We analyze different terms as follows:

$$2(\sqrt{C_r'}\|\boldsymbol{\mu}\| + \sqrt{C_f'}\|\boldsymbol{\mu}\|) \sum_{rf} (\sqrt{C_r'}\|\xi_{r1}\| + \sqrt{C_f'}\|\xi_{f1'}\|) =$$

$$2(\sqrt{C_r'}\|\boldsymbol{\mu}\| + \sqrt{C_f'}\|\boldsymbol{\mu}\|)(n_f \sum_r \sqrt{C_r'}\|\xi_{r1}\| + n_r \sum_f \sqrt{C_f'}\|\xi_{f1'}\|).$$

$$(56)$$

We know that $\|\xi_{r1}\|, \|\xi_{f1'}\|$ are chi-distribution which is also sub-exponential distribution. We can utilize the tail bound for summation of the sub-exponential random variables to obtain high probability bound on the summation. We can have that with probability $2\delta$,

$$2(\sqrt{C_r'}\|\boldsymbol{\mu}\| + \sqrt{C_f'}\|\boldsymbol{\mu}\|) \sum_{rf} (\sqrt{C_r'}\|\xi_{r1}\| + \sqrt{C_f'}\|\xi_{f1'}\|),$$

$$\leq 2(\sqrt{C_r'}\|\boldsymbol{\mu}\| + \sqrt{C_f'}\|\boldsymbol{\mu}\|)(n_r n_f \sqrt{C_r'}\sigma\sqrt{d} + n_f n_r \sqrt{C_f'}\sigma\sqrt{d} + n_f \sqrt{\frac{n_r \sigma^2}{C_1} \log(\frac{2}{\delta})} + n_r \sqrt{\frac{n_f \sigma^2}{C_1} \log(\frac{2}{\delta})}).$$

$$(57)$$

Now, we move to the next chi-square distribution terms $C_r' \sum \|\xi_{r_1}\|^2, C_f' \sum \|\xi_{f_1'}\|^2$. By using the lemma 5.6, we can have that with probability $1 - \delta$,

$$C_r' \sum_{rf} \|\xi_{r_1}\|^2 \leq C_r'(n_f n_r \sigma^2 d + n_f \sqrt{\frac{n_r \sigma^4 d}{C_2} \log(\frac{2}{\delta})}).$$

$$(58)$$

and so is the $C'_f \sum \|\xi_{f'_1}\|^2$,

$$C'_f \sum_{rf} \|\xi_{r_1}\|^2 \le C'_f(n_f n_r \sigma^2 d + n_r \sqrt{\frac{n_f \sigma^4 d}{C_2} \log(\frac{2}{\delta})}). \tag{59}$$

The term $\sum_{rf} \sqrt{C'_r C'_f}\|\xi_{r1}\|\|\xi_{f1'}\|$ can also be dealt with in the same manner,

$$\sqrt{C'_r C'_f} \sum_{rf} \|\xi_{r1}\|\|\xi_{f1'}\| \le \sqrt{C'_r C'_f}(\sum_r \|\xi_{r1}\|)(\sum_f \|\xi_{f1'}\|),$$

$$\le \sqrt{C'_r C'_f}(n_r \sqrt{2}\sigma \frac{\Gamma((d+1)/2)}{\Gamma(d/2)} + \sqrt{\frac{n_r \sigma^2}{C_1} \log(\frac{2}{\delta})})(n_f \sqrt{2}\sigma \frac{\Gamma((d+1)/2)}{\Gamma(d/2)} + \sqrt{\frac{n_f \sigma^2}{C_1} \log(\frac{2}{\delta})}). \tag{60}$$

To simplify the analysis, we only keep terms with magnitude at least $n_f n_r$. We will reach that with probability $1 - 6\delta$

$$\lambda_1(S) \le \mathcal{O}\Big(n_f n_r\big((\sqrt{C'_r} + \sqrt{C'_f})^2 \|\boldsymbol{\mu}\|^2 + 2\sqrt{2}(\sqrt{C'_r} + \sqrt{C'_f})^2 \|\boldsymbol{\mu}\|\sigma(\frac{\Gamma((d+1)/2)}{\Gamma(d/2)})$$
$$+ (C'_r + C'_f)\sigma^2 d + 2\sigma^2 \sqrt{C'_r C'_f}(\frac{\Gamma((d+1)/2)}{\Gamma(d/2)})^2\big)\Big). \tag{61}$$

To see how signal noise ratio (SNR $= \frac{\|\boldsymbol{\mu}\|}{\sigma\sqrt{d}}$) interact with the right hand side, we extract a factor $\sigma^2 d$ from all terms involved:

$$\lambda_1(S) \le \mathcal{O}\Big(n_f n_r \sigma^2 d((\sqrt{C'_r} + \sqrt{C'_f})^2 (\text{SNR})^2 + \frac{2\sqrt{2}}{\sqrt{d}}(\sqrt{C'_r} + \sqrt{C'_f})^2 (\frac{\Gamma((d+1)/2)}{\Gamma(d/2)})(\text{SNR})$$
$$+ (C'_r + C'_f) + \frac{2}{d}\sqrt{C'_r C'_f}(\frac{\Gamma((d+1)/2)}{\Gamma(d/2)})^2)\Big),$$
$$\le \mathcal{O}\Big(n_f n_r \sigma^2 d((\sqrt{C'_r} + \sqrt{C'_f})^2 (\text{SNR})^2 + (C'_r + C'_f))\Big). \tag{62}$$

where in the last equation, we omit terms with d in the denominator as it tends to be large when we consider larger network.

For the second part of the proof, we know that $H_i$ have block structures as follows:

$$\frac{\partial^2 \ell(y_i \cdot f(W, x_i))}{\partial w_{j,r} \partial w_{j',r'}} =$$

$$\ell''_i \cdot \frac{jj'}{m^2} \cdot (\mathbf{1}_{\{\langle w_{j,r}, y_i \cdot \boldsymbol{\mu}\rangle > 0\}} \cdot \boldsymbol{\mu} + \mathbf{1}_{\{\langle w_{j,r}, \xi_i\rangle > 0\}} \cdot y_i \cdot \xi_i)(\mathbf{1}_{\{\langle w_{j',r'}, y_i \cdot \boldsymbol{\mu}\rangle > 0\}} \cdot \boldsymbol{\mu} + \mathbf{1}_{\{\langle w_{j',r'}, \xi_i\rangle > 0\}} \cdot y_i \cdot \xi_i)^T. \tag{63}$$

We can see that the whole $H_i$ matrix can be regarded as outer product of vector $vv^T$ where we have $v_{jr}$ being

$$v_{jr} = \frac{l''_i j}{m}(\mathbf{1}_{\{\langle w_{j,r}, y_i \cdot \boldsymbol{\mu}\rangle > 0\}} \cdot \boldsymbol{\mu} + \mathbf{1}_{\{\langle w_{j,r}, \xi_i\rangle > 0\}} \cdot y_i \cdot \xi_i). \tag{64}$$

We can immediately know that the eigenvalue of the $H_i$ will be upper bounded by $v^T v$ as follows:

$$
\begin{aligned}
\lambda_{\max}(H_i) \leq v^T v &= \frac{l_i''}{m^2} \sum_{jr} (\mathbf{1}_{\{\langle w_{j,r}, y_i \cdot \boldsymbol{\mu}\rangle > 0\}} \cdot \boldsymbol{\mu} + \mathbf{1}_{\{\langle w_{j,r}, \xi_i\rangle > 0\}} \cdot y_i \cdot \xi_i)^2, \\
&\leq 2\frac{l_i''}{m^2} \sum_{jr} \mathbf{1}_{\{\langle w_{j,r}, y_i \cdot \boldsymbol{\mu}\rangle > 0\}} \|\boldsymbol{\mu}\|^2 + \mathbf{1}_{\{\langle w_{j,r}, \xi_i\rangle > 0\}} \|\xi_i\|^2, \\
&\leq 2\frac{l_i''}{m^2} \sum_{jr} \|\boldsymbol{\mu}\|^2 + \|\xi_i\|^2, \\
&\leq 2\frac{1}{m^2} \sum_{jr} \|\boldsymbol{\mu}\|^2 + \|\xi_i\|^2, \\
&= C(\|\boldsymbol{\mu}\|^2 + \|\xi_i\|^2),
\end{aligned}
\tag{65}
$$

where we use $C$ to encompass all constants.

To bound the $\max_{rf} \lambda_{\max}(D_{rf}) = \lambda_{\max}(C_r' H_r + C_f' H_f)$, we can use the following:

$$
\lambda_{\max}(D_{rf}) = \lambda_{\max}(C_r' H_r + C_f' H_f) \leq C_r' \lambda_{\max}(H_r) + C_f' \lambda_{\max}(H_f).
\tag{66}
$$

Then for any $\delta \in (0,1)$, with probability at least $1 - \delta$, we can upper bound the the $H_r$ with the following ($\|\xi_i\|$ is subexponential):

$$
\begin{aligned}
\max_{1 \leq i \leq n_r} C(\|\boldsymbol{\mu}\|^2 + \|\xi_i\|^2) &\leq C\left(\|\boldsymbol{\mu}\|^2 + \sigma^2\Big[d + 2\sqrt{d \log \frac{n_r}{\delta}} + 2\log \frac{n_r}{\delta}\Big]\right), \\
&\leq \mathcal{O}(\|\boldsymbol{\mu}\|^2 + \sigma^2 d).
\end{aligned}
\tag{67}
$$

Similarly, we can have the bound on $H_f$ which is of same order and jointly we can have that with probability $1 - 8\delta$

$$
\lambda_{\max}(D_{rf}) \leq \mathcal{O}((C_r' + C_f')(\|\boldsymbol{\mu}\|^2 + \sigma^2 d)) = \mathcal{O}((C_r' + C_f')\sigma^2 d(\mathrm{SNR}^2 + 1))
\tag{68}
$$

Last is the division and take the limit and we can have the following:

$$
\lim_{\mathrm{SNR}\to 0} \frac{\lambda_{\max}(S)^{\mathrm{upper}}}{\max_{rf} D_{rf}^{\mathrm{upper}}} = \mathcal{O}(n_r n_f) \ , \ \lim_{\mathrm{SNR}\to\infty} \frac{\lambda_{\max}(S)^{\mathrm{upper}}}{\max_{rf} D_{rf}^{\mathrm{upper}}} = \mathcal{O}\left(n_r n_f \left(1 + \frac{2\sqrt{C_r' C_f'}}{C_r' + C_f'}\right)\right).
\tag{69}
$$

$\square$

