# OpenReview forum: "Why Some Models Resist Unlearning: A Linear Stability Perspective"
_ICLR.cc/2026/Conference — Submitted to ICLR 2026_

### Official Review · Reviewer_MYQ6 · 2025-10-26

**Soundness:** 3
**Presentation:** 1
**Contribution:** 3
**Rating:** 4
**Confidence:** 3

**Summary:**

The paper analyzes gradient descent-ascent based machine unlearning, through Linear stability. They provide a theoretical analysis, based on linear stability, that demonstrates when the data coherence between the forget and retain sets prevents machine unlearning, with clear bounds. Subsequently, they also provide experimental evaluations that display this effect of data coherence in machine unlearning.

**Strengths:**

The papers strong point is the theoretical analysis, it is an interesting extension of existing linear stability results to unlearning methods, descent ascent, that require different datasets and where the two datasets are treated differently in the optimization process.

Section 2 provides tight ( by order) bounds for the coherence of the dataset, over the bound the models manage to unlearn and escape the minimum to which the original solution was bound and below which they remain stable around that solution.
The subsequent discussion about the implications of this result for different unlearning methods is also interesting and important

Section 3 provides a very interesting result that correlates the signal to noise ratio with a models ability to move away from the local minimum in which it started. The result provides an interesting observation between data correlations and a models ability to unlearn.

**Weaknesses:**

My main concerns are the following:

1. A primary concern is the general phrasing for the result. The result to the best of my knowledge refers to ascent based machine unlearning, where gradient ascent is applied to the forget set. This is an important distinction from other unlearning methods that don't utilize this technique, such as [1],[2],[3] and as a result I believe some rephrasing in the introduction and related works is important.

2. For related works, [4] also does an analysis for machine unlearning, where the authors discuss similar concerns for Ascent based machine unlearning and showcase in a different setting that correlation impacts machine unlearning. The two works rely on different tools and establish different results.

2. The references don't work, in the version of the pdf that is uploaded, which is a huge issue, as there is no link from the text to the corresponding reference. This is strange as in the iclr template references by default work.

3. In line 151 there is a typo and $f_r$ should be $n_r$

4. In line 260 there is an e missing.

[1] Rewind-to-Delete: Certified Machine Unlearning for Nonconvex Functions  ,Siqiao Mu, Diego Klabjan

[2] Langevin Unlearning: A New Perspective of Noisy Gradient Descent for Machine Unlearning  Eli Chien, Haoyu Wang, Ziang Chen, Pan Li

[3] Attribute-to-Delete: Machine Unlearning via Datamodel Matching Kristian Georgiev, Roy Rinberg, Sung Min Park, Shivam Garg, Andrew Ilyas, Aleksander Madry, Seth Neel

[4] Ascent Fails to Forget Ioannis Mavrothalassitis, Pol Puigdemont, Noam Itzhak Levi, Volkan Cevher

**Questions:**

My primary question has to do with the linear stability of SGD analysis itself. In lines 127 and 128 you utilize a first order Taylor expansion and eliminate the constant factor with respect to $\delta$ due to the fact that $w^* $ is a local minimum of the loss function $L$ and as a result $\nabla L(w^*)=0$. In line 137 you do the same for the stochastic gradient $g_t$, which would correspond to sample $k$ from the dataset. Now in general it is not necessary for the sample to have a gradient that is zero on the local minimum, however in equation (1), line 139 you take it to be zero. I wanted to ask. Am I missing something? Is there an additional assumption that is not stated there?

I only have the question above which is crucial as all of the results rely on this.

---

> ### Author Response · Authors · 2025-11-21
>
> 1. A primary concern is the general phrasing for the result. The result to the best of my knowledge refers to ascent based machine unlearning, where gradient ascent is applied to the forget set. This is an important distinction from other unlearning methods that don't utilize this technique, such as [1],[2],[3] and as a result I believe some rephrasing in the introduction and related works is important.
>
> We thank the reviewer for this helpful observation. We will revise the introduction and related work sections to better position our paper within the broader unlearning literature, including differential privacy based approaches. Our focus, however, is deliberately on gradient-based unlearning methods, where forgetting is realized through optimization dynamics (e.g., gradient ascent on the forget set). Our contribution lies in characterizing how data geometry and curvature jointly determine the success or failure of such methods. We will clarify this scope and include additional references in the revised version.
>
> 2. For related works, [4] also does an analysis for machine unlearning, where the authors discuss similar concerns for Ascent based machine unlearning and showcase in a different setting that correlation impacts machine unlearning. The two works rely on different tools and establish different results.
>
> [4] Ascent Fails to Forget. Ioannis Mavrothalassitis, Pol Puigdemont, Noam Itzhak Levi, Volkan Cevher
>
> We appreciate the reviewer for pointing us to this related work. We view Ascent Fails to Forget [4] as complementary to our study. While both works investigate gradient-based unlearning, they differ substantially in modeling assumptions, analytical scope, and objectives.
>
> The focus on linear logistic regression to understand the relationship between forget set and retain set under gradient descent method (without stochasticity involved). To describe the relationship between retain set and forget set, they assign data with specific structure to transfer the problem effectively into 2 dimensional regime as shown in their proof. The relationship is described through a scalar eplison indicating the overlapping outside its own dataset. Their work focus more on discussion of relationship between the solutions when performing different unlearning methods.
>
> There are several things that we find ourselves distinct from them.
>
> First is that our work did not restricted to linear logistic regression as our work use the loss landscape as foundation to describe the dynamics.
>
> Second, we analyze the optimization behavior under stochasticity in sampling in which they obtain their result through descent or ascent on whole assigned dataset as shown in their formulation and this demonstrates different side of the optimization.
>
> Third, for the diverging criterion and matching lower bound, we did not impose specific data structure design (ex: x = [1, 0, 0, ..]). Our coherence based analysis offer insight for data that is not within this category.
>
> Fourth, their work capture the relationship between forget set and retain set with 2 dimensional entanglement through a scalar eplison. As our work did not assume commute properties of Hessian between samples, we can capture entanglement within high dimensional structure which integrated the potential complicated interaction between sample from retain set and forget set.
>
> Last, we describe the relationship between forgetting and memorization which is not done in their work. The distinction results from the fact that linear logistic regression cannot reach interpolation regime as usual deep learning does.(it cannot fit the data perfectly) Our work analyze the 2 layer ReLU CNN which can achieve this scope and is able to describe this phenomenon in a natural manner.
>
> Taken together, our framework complements [4] by extending the analysis to stochastic and high dimensional entanglement regimes and by providing a geometric explanation of why memorization facilitates forgetting. We believe these perspectives jointly enrich the theoretical understanding of machine unlearning.
>
> 3. The references don't work, in the version of the pdf that is uploaded, which is a huge issue, as there is no link from the text to the corresponding reference. This is strange as in the iclr template reference
>
> We thank the reviewer for catching this formatting issue. We will update the current version to ensure that all references are correctly linked and functioning within the ICLR template.

---

> > ### Author Response · Authors · 2025-11-21
> >
> > 4. My primary question has to do with the linear stability of SGD analysis itself. In lines 127 and 128 you utilize a first order Taylor expansion and eliminate the constant factor with respect to  due to the fact that  is a local minimum of the loss function  and as a result. In line 137 you do the same for the stochastic gradient , which would correspond to sample from the dataset. Now in general it is not necessary for the sample to have a gradient that is zero on the local minimum, however in equation (1), line 139 you take it to be zero. I wanted to ask. Am I missing something? Is there an additional assumption that is not stated there?I only have the question above which is crucial as all of the results rely on this.
> >
> > We thank the reviewer for this insightful question. Our analysis follows the standard linear interpolating regime used in prior works such as [1, 2] widely observed in overparameterized networks, where training loss and sample-wise losses approach zero near convergence. Empirical and theoretical studies [3, 4] have shown that modern neural networks can fit all training labels exactly, implying that the close-to-zero loss results from vanishing per-sample gradients rather than balanced large gradients. In this regime, the local dynamics are dominated by the Hessian curvature, making the linear approximation around minima both standard and justified. We will make this assumption explicit in the revision to avoid confusion.
> >
> > [1] Dexter et al., A Precise Characterization of SGD Stability Using Loss Surface Geometry, ICLR 2024.
> >
> > [2] Lei Wu and Weijie J. Su. The implicit regularization of dynamical stability in stochastic gradient descent. International Conference on Machine Learning, 2023.
> >
> > [3] Tang et al., DP-AdamBC: Your DP-Adam Is Actually DP-SGD (Unless You Apply Bias Correction), AAAI 2024.
> >
> > [4] Chizat and Bach, Implicit Bias of Gradient Descent for Wide Two-Layer Neural Networks Trained with the Logistic Loss, COLT 2020.

---

> > > ### Comment · Reviewer_MYQ6 · 2025-11-24
> > >
> > > I would like to thank the authors for the extensive response. While the authors have promised to resolve the concerns raised 1->3 in an updated version of their work, I would like to point out that iclr offers the authors the possibility to update their manuscript during the rebuttal. So I would request to see the updated version before making any changes to my evaluation.
> > >
> > > I would also like to point out that as I stated in my original review the theoretical assumptions in the work are logical from my standpoint and I fully agree with the general comment of the authors. However, when a work is theoretical a clear statement of all of the assumptions made, with a relevant section maybe in the appendix to explain the significance of these assumptions, is extremely important to prevent misleading people reading the work. So I would request that in a revised manuscript the assumptions such as the ones made clear from your response in 4 are stated clearly. A general term for this assumption is relative noise that is indeed common in the literature.

---

> > > > ### Author Response · Authors · 2025-11-25
> > > >
> > > > We thank the reviewer for the thoughtful feedback. We have updated the manuscript to incorporate all points raised during the discussion, and we have added clarifications, additional experiments, and new supporting analyses as requested. We hope the revision addresses the reviewer’s concerns, and we would be grateful for any further comments or suggestions if the reviewer finds them necessary.

---

### Official Review · Reviewer_j7DZ · 2025-10-29

**Soundness:** 2
**Presentation:** 2
**Contribution:** 3
**Rating:** 4
**Confidence:** 2

**Summary:**

This paper discusses the relation between memorization of noise and coherence across samples as they relate to unlearning.  It developed a theoretical framework based on linear stability under which standard unlearning techniques would diverge/converge.  It then conducted an experimental study to observe the link between forgetting, coherence, and memorization.

**Strengths:**

Strong points of the paper include the soundness of the framework, innovation towards utilizing linear stability to enhance ML unlearning using forget and retain sets, and relatively easy-to-follow organization.

**Weaknesses:**

The main pitfall of the papers lies in the minimum space allocated to discussing the experiment which is a large portion of the discussed findings for this paper, leading to confusion and currently inconclusive results (as they appear in paper).

There is not enough discussion for the results of the experiment, and the graphs are hard to read when considering what they are trying to convey.  Going more in depth to describe them would be helpful for readers comprehension.

While the main purpose of the experiment is to demonstrate the effects of lower and higher SNR’s, the provided experimental study failed to serve the purpose, since noise appears to be set to a fixed value according to section 3.2.  If this is not the case it would be helpful to go more in depth into how noise is varied between tests.

Readability: some words appear to be placed for the purpose of enhanced vernacular; however, they are confusing at times (due to appearing incorrect) and could be replaced with better words to convey the appropriate idea.  One such example is “more spurious noise.”

**Questions:**

How is noise varied between tests, as in the explanation provided in section 3.2, it appears to be fixed?

---

> ### Author Response · Authors · 2025-11-21
>
> 1. The main pitfall of the papers lies in the minimum space allocated to discussing the experiment which is a large portion of the discussed findings for this paper, leading to confusion and currently inconclusive results (as they appear in paper). There is not enough discussion for the results of the experiment, and the graphs are hard to read when considering what they are trying to convey. Going more in depth to describe them would be helpful for readers comprehension.
>
> We thank the reviewer for the feedback and apologize for the lack of clarity. Below we clarify the design and purpose of our experiments and will add the discussion into the updated version of our work. The experiments serve two distinct goals: (1) verifying the theoretical divergence criterion and matching lower bound under a controlled synthetic setting, and (2) empirically testing the predicted link between memorization and forgetting using a two-layer CNN.
>
> Synthetic verification.
> We construct sample-wise Hessians such that the overall dataset Hessian is constrained by
> 2 / $\eta$, ensuring that any divergence behavior arises purely from stochasticity and data geometry. We vary the sample-wise Hessian coherence while maintaining the same global sharpness, then run training under different learning rates. Divergence is detected when the weight norm grows 1000× larger than initialization (repeated over five runs). As shown in Fig. 1, the empirically observed divergence boundary (red/blue transition) aligns closely with both our theoretical criterion and the matching lower bound, confirming that the theory precisely predicts optimization stability.
>
> Memorization–forgetting test.
> For the second part, we want to verify the whether or not stronger memorization will lead to stronger forgetting or unlearning as our theory indicates that when model memorize the data, it will map the data to space where it is orthogonal to the main dataset in terms of loss curvatures and give low coherence measure. This low coherence measure will therefore lead to easier forgetting process as predicted by our theory. To control memorization strength, we generate datasets with varying signal-to-noise ratio (SNR). Low-SNR data force the model to memorize noise (learn orthogonal noise directions in the loss curvature space) yielding low coherence. After training, we apply our unlearning procedure and track forget losses. The training loss indicates that all models properly learn the data and converge. The testing loss is to demonstrate whether or not the model memorize the data. This is due to the fact that the when the model perform well in train but bad in the testing, it indicate there exist memorization. We therefore can observe that small SNR regime show strong memorization. Lastly, the forget loss indicate that whether or not we can escape the current minima and perform successful unlearning. Our results show that it is consistence with our theoretical prediction. The regime of memorization will also give stronger forgetting results aligning with our coherence measure.
>
> 2. While the main purpose of the experiment is to demonstrate the effects of lower and higher SNR’s, the provided experimental study failed to serve the purpose, since noise appears to be set to a fixed value according to section 3.2. If this is not the case it would be helpful to go more in depth into how noise is varied between tests. How is noise varied between tests, as in the explanation provided in section 3.2, it appears to be fixed?
>
> We apologize for the confusion. The key quantity in our analysis is the signal-to-noise ratio (SNR), which can be adjusted by varying either the signal strength or the noise magnitude while holding the other fixed. In our experiments, we vary the signal strength to achieve different SNR levels (the noise variance remains constant). This is stated in the second paragraph of Section 3.2: “Figure 2 shows heatmaps over signal strength and dimension.” The same can be seen in Figure 2, where the x-axis directly corresponds to signal strength. We will clarify this design choice explicitly in the revised version to avoid ambiguity.
>
> 3. Readability: some words appear to be placed for the purpose of enhanced vernacular; however, they are confusing at times (due to appearing incorrect) and could be replaced with better words to convey the appropriate idea. One such example is “more spurious noise.”
>
> We thank the reviewer for pointing this out. We will refine this statement, and any other the reviewer finds to be unclear.

---

> > ### Author Response · Authors · 2025-11-25
> >
> > We thank the reviewer for the thoughtful feedback. We have updated the manuscript to incorporate all points raised during the discussion, and we have added clarifications, additional experiments, and new supporting analyses as requested. We hope the revision addresses the reviewer’s concerns, and we would be grateful for any further comments or suggestions if the reviewer finds them necessary.

---

### Official Review · Reviewer_G3DN · 2025-10-31

**Soundness:** 3
**Presentation:** 3
**Contribution:** 2
**Rating:** 4
**Confidence:** 4

**Summary:**

This paper introduces a novel and valuable theoretical framework for machine unlearning, grounding the process in linear stability analysis. The core concepts of "data coherence" and the stability frontier are significant contributions. The insight that stronger memorization (via low SNR data) can facilitate unlearning is particularly provocative and insightful. However, the paper's practical relevance is constrained by strong, potentially unrealistic assumptions and an overly simplified experimental validation. While theoretically elegant, the work suffers from a gap between its theoretical claims and the complexities of real-world unlearning scenarios.

**Strengths:**

1. The paper's primary strength is its novel application of linear stability analysis to machine unlearning. Framing unlearning as a problem of escaping a local minimum, rather than standard training, is a powerful conceptual shift. This provides a principled lens to analyze the underlying dynamics, moving the field away from purely empirical observations.
2.  The introduction of "data coherence" as a measure of Hessian alignment is a key contribution. It provides an intuitive and quantifiable way to understand the critical interaction between the "retain" and "forget" sets. This concept elegantly explains why unlearning is difficult when the data to be forgotten is structurally similar to the data being kept.
3. The paper's most impactful finding is the rigorously argued link between memorization and forgettability ("the more you memorize, the easier you forget"). By connecting memorization to low signal-to-noise ratio (SNR) data, which in turn leads to low coherence, the authors establish a surprising and deep connection between unlearning, generalization, and data geometry. This is a significant insight with broad implications.

**Weaknesses:**

1. The Linear Approximation is a Strong Limitation: The entire framework is built upon a local linear approximation of the loss landscape around a minimum (w*). This assumption is fragile in deep learning. The unlearning process, especially when successful (divergent), inherently moves parameters far from this local region, invalidating the approximation. Furthermore, it fails to account for the flat, wide minima common in modern networks where the Hessian may be ill-conditioned.
2. Definition of "Successful Unlearning" is Impractical: The paper equates successful unlearning with the divergence of model parameters (E[||w_k||^2] → ∞). This is a mathematical abstraction that does not align with practical goals. A useful unlearning algorithm must not only forget specific data but also converge to a new state where utility on the retain set is preserved. The current framework provides no guarantees or analysis regarding this crucial aspect.
3.  The central metric relies on analyzing per-sample Hessians. For any large-scale, modern neural network, computing these Hessians is computationally infeasible. This severely limits the theory's applicability as a practical diagnostic or predictive tool, rendering it more explanatory than prescriptive.
4. The validation is performed on a synthetic dataset with a simple two-layer CNN. This "toy" setup, while clean for validating the theory, leaves a significant open question about whether the findings generalize to complex, high-dimensional datasets (e.g., ImageNet) and deep architectures (e.g., ResNets, Transformers).
5. The experiments confirm the paper's theoretical bounds but do not benchmark against any state-of-the-art (SOTA) approximate unlearning algorithms. It is unclear how the dynamics predicted by this theory relate to the practical performance (e.g., efficacy, efficiency, utility preservation) of widely-used unlearning methods.
6. The key finding—"the more you memorize, the easier you forget"—is derived and tested under a specific signal-plus-noise data model. This is a narrow definition of memorization. It is not clear if this insight holds for other forms of memorization, such as the verbatim memorization of sequences in LLMs, which may arise from different mechanisms.

**Questions:**

1.	Acknowledge the limitations of the linear stability assumption and clarify that the theory is best suited to explain the onset of divergence rather than the entire unlearning process.
2.	 Provide at least preliminary experiments on a more realistic benchmark (e.g., ResNet on CIFAR-10) to demonstrate if the theoretical predictions hold in more complex settings.
3.	Discuss how the theoretical condition of divergence could be connected to practical unlearning goals, such as membership inference attack success or the preservation of retain set accuracy.

---

> ### Author Response · Authors · 2025-11-21
>
> 1. The Linear Approximation is a Strong Limitation. The unlearning process inherently moves parameters far from this local region. It fails to account for the flat, wide minima common in modern networks where the Hessian may be ill-conditioned.
>
> We thank the reviewer for highlighting this important point. We agree that the linear (quadratic) approximation introduces a limitation. This is true for *any* theoretical work that studies neural networks (gradient flows, NTK-based, or limited to just logistic regression settings as the just published Neurips25 paper "Ascent Fails to Forget" pointed out by the reviewer MYQ6 etc). Our focus on linear stability is grounded in the fact that it has been extensively and successfully used in theoretical analyses of deep learning dynamics[1][2][3][4], providing insights that align well with empirical observations[5][6][7]. Empirical studies have also shown that the loss landscape near minima is often well-approximated by a quadratic form, supporting the validity of this assumption for characterizing local stability.
>
> In our setting, the assumption is particularly reasonable because the unlearning process begins from a pre-trained minimum and proceeds via small fine-tuning steps. Our framework therefore focuses on divergence and the transition out of stability. For wide or flat minima, our analysis only requires the Hessian to be positive semi-definite. No strong-convexity or conditioning assumptions are imposed. This suffices for deriving the matching lower bound and divergence criterion (those are bounds shown in the experiments) and results regarding memorization and forgetting, which constitute our main contributions.
>
> [1] Dexter et al., A Precise Characterization of SGD Stability Using Loss Surface Geometry, ICLR 2024.
>
> [2] Lei Wu and Weijie J. Su. The implicit regularization of dynamical stability in stochastic gradient descent. International Conference on Machine Learning, 2023.
>
> [3] Arseniy Andreyev. “Edge of Stochastic Stability: Revisiting the Edge of Stability for SGD”. arXiv preprint arXiv:2412.20553 (2025).
>
> [4] WEI-KAI CHANG and Rajiv Khanna, A Unified Stability Analysis of {SAM} vs {SGD}: Role of Data Coherence and Emergence of Simplicity Bias, Neural Information Processing Systems, 2025
>
> [5] Hao Li. Visualizing the loss landscape of neural nets. Advances in Neural Information Processing Systems (NeurIPS), 2018. https://arxiv.org/abs/1712.09913.
>
> [6] Sidak Pal Singh. Woodfisher: Efficient second-order approximation for neural network compression. In NeurIPS, 2020
>
> [7] Behrooz Ghorbani. In International Conference on Machine Learning, PMLR,2019.
>
> 2. Definition of "Successful Unlearning" is Impractical.
>
> The reviewer seems to have misunderstood. In our framework, we interpret the divergence as a mathematical indicator of initiation of forgetting. Specifically, our divergence criterion characterizes the boundary where the system leaves the basin of the original solution and begins searching for a new equilibrium **consistent with the retain set, and not just to forget specific forget-data**. Further more, the norm and distance can be effectively transfer to loss which is widely reported in unlearning work to characterize different algorithms [1][2][3][4][5][6]. Also, the distance between original solution and fine-tuned solution is also one object to analyze in theoretical framework[7]. All of these together show that this is still a useful quantity that is valuable for further investigation and to understand the unlearning problem through optimization perspective.
>
> This lens provides a explanation of the mechanistic difficulty of unlearning in gradient-based methods and reveals how the coupling between forget and retain curvatures governs that transition. We view this as a foundational step toward future analyses that will model the post-divergence dynamics and utility preservation explicitly and serve as one valuable piece for the community.
>
> [1] C. Fan, J. Liu, Y. Zhang, D. Wei, E. Wong, and S. Liu, “Salun: Empowering machine unlearning via gradient-based weight saliency in both image classification and generation,” ArXiv,2023.
>
> [2] Meghdad Kurmanji, Peter Triantafillou, and Eleni Triantafillou. Towards unbounded machine
> unlearning. NeurIPS, 2023.
>
> [3] Nathaniel Li. The WMDP Benchmark: Measuring and Reducing Malicious Use With Unlearning. arXiv preprint arXiv, 2024.
>
> [4] Laura Graves. Amnesiac machine learning. In Proceedings of
> the AAAI Conference on Artificial Intelligence, 2021.
>
> [5] Aditya Golatkar. Eternal sunshine of the spotless net: Selective forgetting in deep networks. In Proceedings of the IEEE/CVF Conference on Computer Vision and Pattern Recognition, 2020.
>
> [6] Anvith Thud. Unrolling sgd: Understanding factors influencing machine unlearning. In 2022 IEEE 7th European Symposium on Security and Privacy (EuroSP), pages 303–319. IEEE, 2022.
>
> [7] , Pol Puigdemont, Ascent Fails to Forget Ioannis Mavrothalassitis Nips 2025

---

> > ### Author Response · Authors · 2025-11-21
> >
> > 3. The central metric relies on analyzing per-sample Hessians. For any large-scale, modern neural network, computing these Hessians is computationally infeasible. The validation is performed on a synthetic dataset with a  simple two-layer CNN. This "toy" setup, while clean for validating the theory, leaves a significant open question about whether the findings generalize to complex, high-dimensional datasets (e.g., ImageNet) and deep architectures (e.g., ResNets, Transformers).
> >
> > Our use of Hessian alignment is consistent with other current literature on theoretical studies of linear stability[1][2][3]. We agree that computing per-sample Hessians is computationally expensive for large-scale networks. Our goal, however, is to introduce a conceptual and theoretical framework for understanding the geometry of unlearning, rather than to propose a directly deployable diagnostic tool at this stage. Many foundational quantities in deep learning such as the Hessian itself, the Fisher Information Matrix, and the Neural Tangent Kernel were initially intractable to compute exactly but later inspired efficient approximations and practical algorithms. For example, the Hessian inspired a wide range of curvature-based optimizers such as Sharpness-Aware Minimization (SAM) and Eigenvalue Regularization. The Neural Tangent Kernel (NTK) framework motivated scalable linearization analyses and kernel-based training diagnostics.
> >
> > Similarly, our coherence-based formulation provides an analytic lens that can guide future approximations and practical criteria for unlearning dynamics. As an early step, we validated the theoretical bounds in a controlled setting, where the Hessians can be computed precisely, to ensure clarity of interpretation. We view extending these ideas to scalable approximations and complex architectures (e.g., ResNets, Transformers) as an exciting and natural direction for future work.
> >
> > [1] R. Mulayoff. Exact mean square linear stability analysis for sgd. In Conference on Learning Theory (COLT), 2024.
> >
> > [2] Lei Wu and Weijie J. Su. The implicit regularization of dynamical stability in stochastic gradient descent. International Conference on Machine Learning, 2023.
> >
> > [3] G. Dexter. A precise characterization of sgd stability using loss surface geometry. ICLR, 2024.
> >
> > 4. The experiments confirm the paper's theoretical bounds but do not benchmark against any approximate unlearning algorithms. It is unclear how the dynamics predicted by this theory relate to the practical performance  of widely-used unlearning methods.
> >
> > We have already presented this in Section 2.3. For consistency, we summarize the explanation here.
> >
> > Naive negative-gradient.[1][2]
> > A simple baseline is gradient ascent on the forget set alone. Our framework explains its failure: when the forget set has high internal coherence, its gradients align with the local curvature at $w^*$, so ascent moves along a stable direction that cannot effectively escape the minimum with small learning. Divergence is therefore suppressed, consistent with reports that such methods stagnate or oscillation degrading retain performance while barely reducing forget accuracy.
> >
> > Random-label perturbation.[3]
> > Randomizing labels or injecting noise into the forget gradients effectively \emph{breaks coherence}. The resulting uncorrelated gradients lower the forget-set $\sigma$, enabling faster escape from the original basin. Because these noisy directions are nearly orthogonal to the retain-set curvature, cross-coherence also decreases. Hence, random-label methods succeed by deliberately reducing $\sigma$ in the forget set so that the divergence criterion is more easily met.
> >
> > Min--Max (targeted-forget) methods.[4][5][6]
> > More advanced approaches apply ascent/descent only in directions most associated with the forget set which often use projection operators $P_F$ and $P_R$ on the respective Hessians. These projections isolate subspaces of influence, reducing overlap between $H_f$ and $H_r$ and thus lowering cross-coherence. In our framework, this corresponds to a smaller $\sigma$ and more effective unlearning, as retain and forget dynamics become disentangled.
> >
> > These mappings demonstrate that our theory complements existing algorithmic approaches by explaining why certain strategies succeed (those that reduce coherence) and why others stagnate (those trapped by high coherence). Extending this analysis into other algorithmic benchmarks is a promising next step for future work.
> >
> > [1]Meng Ding. Understanding fine-tuning in approximate unlearning: A theoretical perspective, 2025.
> >
> > [2]Xindi Fan. Imu: Influence-guided machine unlearning, 2025.
> >
> > [3]Laura Graves. Amnesiac machine learning, 2020.
> >
> > [4]Haoran Tang and Rajiv Khanna. Sharpness-aware machine unlearning, 2025, arxiv.
> >
> > [5]Chongyu Fan. Salun: Empowering machine unlearning via gradient-based weight saliency in both image classification and
> > generation, 2024, arxiv.
> >
> > [6] Meghdad. Towards unbounded machine unlearning. NeurIPS, 2023.

---

> > > ### Author Response · Authors · 2025-11-21
> > >
> > > 5. The key finding "the more you memorize, the easier you forget" is derived and tested under a specific signal-plus-noise data model. This is a narrow definition of memorization. It is not clear if this insight holds for other forms of memorization, such as the verbatim memorization of sequences in LLMs, which may arise from different mechanisms.
> > >
> > > Our definition of memorization is grounded in the observation that, to minimize training loss, models often overfit to highly orthogonal components of the data and the directions that are uncorrelated with the main signal. This view aligns with our coherence-based analysis: in our signal-plus-noise experiments, noise components are orthogonal in expectation, and the theory predicts that such directions are easily forgotten once ascent begins.
> > >
> > > Similar notions have been explored in recent work. [1] show that memorized examples correspond to orthogonal activation patterns within the network, which translate into orthogonal Hessian directions, while [2] study memorization in highly orthogonal subspaces. These results support our geometric interpretation that memorization arises from localized, low-coherence modes.
> > >
> > > The type of memorization captured by our coherence framework (fitting orthogonal directions or outlier features) is one of the most fundamental and widely studied forms of memorization in modern deep learning. It is closely connected to optimization stability, generalization behavior, and ultimately forgetting. We agree that verbatim sequence memorization in large language models may involve additional mechanisms; however, the underlying causes of such memorization remain an open research question with no corresponding theoretical formulation and/or studies to the best of our knowledge yet. Because our work focuses on the optimization geometry governing gradient-based learning, extending coherence-based analysis to sequential or long-context memorization in LLMs represents an exciting future direction.
> > >
> > > [1] Kaiyue Wen, Tengyu Ma, and Zhiyuan Li. Sharpness minimization algorithms do not only minimize sharpness to achieve better generalization. In Proc. Adv. Neural Info. Processing Systems, volume 36, 2023b.
> > >
> > > [2] Lijia Yu, Xiao-Shan Gao, Lijun Zhang, and Yibo Miao. Generalizability of memorization neural networks. arXiv preprint arXiv:2411.00372, 2024.
> > >
> > > 6. Acknowledge the limitations of the linear stability assumption and clarify that the theory is best suited to explain the onset of divergence rather than the entire unlearning process.
> > >
> > > Any theoretical work on deep learning does make some simplifying assumptions (e.g. NTK-based analysis, infinite-width analysis, gradient flows or limited to just logistic regression settings as the just published Neurips25 paper "Ascent Fails to Forget" pointed out by the reviewer MYQ6 etc), and we agree that our linear stability assumption is one restriction of our work. We have acknowledged it in the title itself. However, our goal is precisely to analyze the initiation of divergence and the transition point where the system becomes unstable and begins to move away from the original solution. We will clarify this scope in the revision to emphasize that the framework provides theoretical insight into when unlearning becomes possible, complementing empirical studies that examine how the model evolves afterward.

---

> > > > ### Author Response · Authors · 2025-11-21
> > > >
> > > > 7. Provide at least preliminary experiments on a more realistic benchmark (e.g., ResNet on CIFAR-10) to demonstrate if the theoretical predictions hold in more complex settings.
> > > >
> > > > We thank the reviewer for this suggestion. We have conducted additional experiments on a more realistic setting: CIFAR-10 with a ResNet-18 model. We first train the model to convergence (100 percent training accuracy with loss less than 0.002). Afterward, we perform unlearning algorithm as stated in main context. We use step size with 0.01 with forget set being 10 percent of the training set. The alpha or weighting between forgetset and training set is set to 0.3. We record the loss on the forget set for the first 500 steps with 50 steps checking points.To probe the relationship between memorization and unlearning predicted by our theory, we inject Gaussian noise of varying variance (0.1,0.3,0.5) into the inputs. Higher noise variance produces stronger memorization, since the network overfits the idiosyncratic noise patterns. As predicted by our coherence framework, models with higher memorization (larger variance) exhibit \emph{faster loss increase on the forget set} during unlearning.
> > > >
> > > > |   step | variance=0.1    | variance=0.3    | variance=0.5    |
> > > > |-------:|:----------------|:----------------|:----------------|
> > > > |      0 | 0.0016 ± 0.0005 | 0.0019 ± 0.0004 | 0.0016 ± 0.0004 |
> > > > |     50 | 0.0024 ± 0.0014 | 0.0016 ± 0.0003 | 0.07 ± 0.073    |
> > > > |    100 | 0.0694 ± 0.0858 | 0.0428 ± 0.0539 | 0.0715 ± 0.0667 |
> > > > |    150 | 0.0477 ± 0.0508 | 0.0356 ± 0.0401 | 0.1575 ± 0.1692 |
> > > > |    200 | 0.0722 ± 0.0762 | 0.1582 ± 0.1744 | 0.6279 ± 0.5556 |
> > > > |    250 | 0.1758 ± 0.2054 | 0.1671 ± 0.1508 | 1.1366 ± 0.6084 |
> > > > |    300 | 0.3888 ± 0.4347 | 0.5626 ± 0.5598 | 1.4868 ± 0.9849 |
> > > > |    350 | 0.6515 ± 0.7707 | 0.8544 ± 0.7993 | 2.4464 ± 1.5124 |
> > > > |    400 | 1.4362 ± 1.5605 | 2.3895 ± 1.2796 | 5.5148 ± 1.8706 |
> > > > |    450 | 2.5029 ± 2.3457 | 2.9324 ± 1.2882 | 5.5811 ± 1.8397 |
> > > >
> > > > 8. Discuss how the theoretical condition of divergence could be connected to practical unlearning goals, such as membership inference attack success or the preservation of retain set accuracy.
> > > >
> > > > We thank the reviewer for this insightful question. At the end of training (or before unlearning), the model typically resides at a solution shaped jointly by both the forget and retain sets. Effective unlearning requires initiating a drift away from this equilibrium and remove the residual influence of the forget set while maintaining stability on the retain set.
> > > >
> > > > Our divergence criterion quantifies the difficulty of initiating this drift through coherence analysis: when forget and retain directions are highly aligned, the model remains trapped in a stable basin, making it hard for any gradient-based method to erase forget-set influence. This directly connects to practical unlearning goals. If an algorithm cannot escape this stable region, it will struggle to reduce membership-inference success (since forget samples remain predictable). Thus, our framework explains a fundamental limit in gradient-based unlearning. Extending the geometric perspective of stability in loss landscapes to predict quantitative changes in these MIA and other privacy metrics is an open and promising direction for future work.

---

> > > > > ### Author Response · Authors · 2025-11-25
> > > > >
> > > > > We thank the reviewer for the thoughtful feedback. We have updated the manuscript to incorporate all points raised during the discussion, and we have added clarifications, additional experiments, and new supporting analyses as requested. We hope the revision addresses the reviewer’s concerns, and we would be grateful for any further comments or suggestions if the reviewer finds them necessary.

---

### Official Review · Reviewer_3vnF · 2025-11-02

**Soundness:** 3
**Presentation:** 1
**Contribution:** 3
**Rating:** 4
**Confidence:** 3

**Summary:**

This paper aims to understand the unlearning guarantees of the unlearning algorithms based on gradient ascent. In particular, there are many unlearning algorithm that operates based on the following update rule

w_{k+1} = w_k - η * ( (1-α) * (1/B) * sum_{i in Br_k} ∇ℓ_i(w_k) - α * (1/B) * sum_{j in Bf_k} ∇ℓ_j(w_k) )

Then, the authors use linear approximation for the gradient which implies that g_k approx H_k w_k. Using this approximation it is easy to obtain a closed-form expression for the update rule given above. Then, the authors define a measure of unlearning which is based on having ||w_t|| to infinity or ||w_t|| becomes constant. The intuition is that if ||w_t|| becomes infinity it means that the model can escape from the minima that it learnt from both train set and forget set.

Their main contribution is a measure that predicts whether ||w_t|| is staying constant or it diverges . Their “unlearning coherence” is a scalar that quantifies how aligned the retain and forget curvature directions are near the trained solution. Intuitively, high unlearning coherence means retain and forget curvatures point in similar directions, so ascent on forget is canceled by descent on retain and the updates tend to stay near the old minimum (harder to “unlearn” in their stability sense); low σ means the directions are incoherent, cancellation is weak, and the iterates can escape the old minimum more easily.

**Strengths:**

I think the perspective provided in this work seems very interesting and it would lead to a prescriptive theory related to unlearning using gradient ascent and descent. I think the unlearning coherence metric that explains when ascent on forget is neutralized by descent on retain seems an actionable metric.

**Weaknesses:**

I am not fully convinced with the definition of unlearning proposed in the paper. Consider a scenario where all per-example gradients (and effectively the per-sample curvatures) are aligned; adding or removing a sample doesn’t change the optimization path or the final solution, so by a deletion standard the model is already “unlearned” without any unlearning step, yet the paper’s coherence lens would label this case as resistant (high coherence ⇒ no escape), which contradicts the intended notion of unlearning—this exposes a scope mismatch between “ability/need to move under mixed ascent/descent” and “deletion is satisfied.”

A second weakness is that success is proxied by divergence of ||w_t|| (and, empirically, by forget-loss increases) rather than by deletion equivalence to retraining. I think the right empirical metric is membership inference based test. I don't think forget-loss increase is an interesting measure.

**Questions:**

please discuss the weakness raised above.

---

> ### Author Response · Authors · 2025-11-21
>
> 1. I am not fully convinced with the definition of unlearning proposed in the paper. Consider a scenario where all per-example gradients (and effectively the per-sample curvatures) are aligned; adding or removing a sample doesn’t change the optimization path or the final solution, so by a deletion standard the model is already “unlearned” without any unlearning step, yet the paper’s coherence lens would label this case as resistant (high coherence means no escape), which contradicts the intended notion of unlearning—this exposes a scope mismatch between “ability/need to move under mixed ascent/descent” and “deletion is satisfied.”
>
> We thank the reviewer for raising this insightful point. We understand the reviewer’s example: if all per-sample gradients (and thus curvatures) are perfectly aligned, removing one sample does not affect the optimization trajectory—by a deletion-based definition, the model is trivially “unlearned.”
>
> To make this concrete, consider an extreme case where the same data point appears in both the retain and forget sets. Removing it would clearly change nothing; by the deletion criterion, this sample is already unlearned. We understand this definition of unlearning is also used in practice in some papers such as unlearning of clusters in federated settings[1]. However, if we look at the problem from other viewpoint that is also often used in practice[2][3], we ask is it possible to undo the effect of the sample brought to the system? Our answer is no since learning the other will automatically incur learning of another copy. The forget sample contributes no unique information. Our framework focuses on the non-trivial regime: when the forget sample has a distinct influence but is dynamically coupled with the retain set. In such cases, gradient-based unlearning cannot succeed without escaping the current basin.
>
> [1] Y. Tao, “Communication efficient and provable federated unlearning,” 2024, arXiv:2401.11018.
>
> [2] Laura Graves. Amnesiac machine learning. In Proceedings of the AAAI Conference on Artificial Intelligence, 2021.
>
> [3] Aditya Golatkar. Eternal sunshine of the spotless net: Selective forgetting in deep networks. In Proceedings of the IEEE/CVF Conference on Computer Vision and Pattern Recognition, 2020.
>
> 2. A second weakness is that success is proxied by divergence of $||w_t||$ (and, empirically, by forget-loss increases) rather than by deletion equivalence to retraining. I think the right empirical metric is membership inference based test. I don't think forget-loss increase is an interesting measure.
>
> We thank the reviewer for this valuable comment. We agree that membership-inference-attack (MIA) metrics provide an important privacy-based evaluation of unlearning. Memberships Inference is just one kind of attack, and there are many variants of it. However, the field does not currently have a single gold-standard metric—different application domains emphasize different notions of “successful unlearning,” such as bias removal, confusion resolution, or privacy protection. Our use of the metric is consistent with many established works that continue to evaluate unlearning effectiveness using forget-set loss or accuracy, which serve as optimization-level proxies for forgetting. Examples include forgetting loss [4, 5], forgetting error [2], forget performance [3], unlearning accuracy [1], and distance-based measures [6]. They are also work characterizing the distance between solutions resulting from different algorithms [7].
>
> Our goal is to characterize the fundamental relation between retain and forget dynamics within gradient-based optimization. From this perspective, loss (and implicitly the weight dynamics it reflects) provides a natural and widely interpretable language to study unlearning behavior (also used in many works). We will clarify this scope in the revised manuscript.
>
> [1] C. Fan, “Salun: Empowering machine unlearning via gradient-based weight saliency in both image classification and generation,” ArXiv,2023.
>
> [2] Meghdad Kurmanji. Towards unbounded machine unlearning. NeurIPS, 2023.
>
> [3] Nathaniel Li. The WMDP Benchmark: Measuring and Reducing Malicious Use With Unlearning. arXiv preprint arXiv, 2024.
>
> [4] Laura Graves. Amnesiac machine learning. In Proceedings of the AAAI Conference on Artificial Intelligence, 2021.
>
> [5] Aditya Golatkar. Eternal sunshine of the spotless net: Selective forgetting in deep networks. In Proceedings of the IEEE/CVF Conference on Computer Vision and Pattern Recognition, 2020.
>
> [6] Anvith Thudi. Unrolling sgd: Understanding factors influencing machine unlearning. In 2022 IEEE 7th European Symposium on Security and Privacy (EuroSP), pages 303–319. IEEE, 2022.
>
> [7] Pol Puigdemont, Ascent Fails to Forget Ioannis Mavrothalassitis,  Nips 2025

---

> > ### Author Response · Authors · 2025-11-25
> >
> > We thank the reviewer for the thoughtful feedback. We have updated the manuscript to incorporate all points raised during the discussion, and we have added clarifications, additional experiments, and new supporting analyses as requested. We hope the revision addresses the reviewer’s concerns, and we would be grateful for any further comments or suggestions if the reviewer finds them necessary.

---

### Author Response · Authors · 2025-11-22
**Responses for all reviewers**

1. Assumption of the theory and its limitation.

Several reviewers raised concerns regarding the quadratic approximation in our analysis. We appreciate this concern, and here we provide a consolidated and detailed justification for why this modeling choice is (1) standard, (2) empirically grounded, and (3) necessary for theoretical progress in unlearning.

First, the local quadratic approximation is a well-established and widely validated modeling framework in the theory of deep learning. A large body of recent work successfully explains diverse optimization behaviors using this approximation, including the Edge-of-Stability phenomenon[1,2], implicit bias and generalization[3,4], eigenvalue dynamics and curvature structures[5], and stability versus divergence of SGD[6,7]. While the approximation does impose limitations as (all theoretical frameworks do) it also provides meaningful insights into real-world systems that alternative approaches currently cannot offer. The success of these works illustrates that quadratic modeling is both theoretically fruitful and empirically relevant. Use of the same approximation follows this tradition and enables us to reveal new connections between the forget set and retain set via their geometric interactions.

Second, multiple empirical studies show that the loss landscape near well-trained minima is smooth and locally well-approximated by a quadratic form[8,9,10]. This has been repeatedly observed across architectures, datasets, and training regimes. Importantly, unlearning is a fine-tuning procedure that begins from an already-converged minimum, precisely the region where such local approximations are the most accurate. Thus, the theoretical foundation is not only supported by prior empirical evidence but is also particularly appropriate for modeling unlearning dynamics.

Third, a key question is: why do we need approximations at all in theoretical unlearning?
Existing theoretical works on machine unlearning also rely on simplified or linearized systems in order to obtain analyzable results. For example, [11] study unlearning under a quadratic model and analyze the drift of optimization trajectories which is exactly the type of measure we employ. [12] use linear feature–weight dot product models and weight-space distances to characterize approximate unlearning. The recent work [13] operates in linear logistic regression, enabling closed-form solutions and a theoretical characterization of forgetting difficulty. [14] analyze unlearning by unrolling the SGD recursion, which again relies on linearization of the gradient dynamics.

One commonality across all these theoretical frameworks is that they require simplifying assumptions to make the problem analyzable. Our quadratic approximation fits within this established methodology. Although no single abstraction captures the full complexity of deep networks, each sheds light on a different aspect of the phenomenon. Our framework contributes by offering a principled geometric description of how the retain and forget sets interact through curvature and coherence, revealing a new mechanistic explanation for when gradient-based unlearning becomes difficult or feasible.

[1] Arseniy Andreyev. “Edge of Stochastic Stability: Revisiting the Edge of Stability for SGD”. arXiv, 2025.

[2] Lee, S. and Jang, C. A new characterization of the edge of stability based on a sharpness measure aware of batch gradient distribution. ICLR, 2023.

[3] Lei Wu and Weijie J. Su. The implicit regularization of dynamical stability in stochastic gradient descent. ICLR, 2023.

[4] Lei Wu, Mingze Wang, and Weijie Su. The alignment property of sgd noise and how it helps select flat minima: A stability analysis. Advances in Neural Information Processing Systems, 2022.

[5] Atish Agarwala. Sam operates far from home: eigenvalue regularization as a dynamical phenomenon. arXiv preprint arXiv:2302.08692, 2023.

[6] Dexter et al., A Precise Characterization of SGD Stability Using Loss Surface Geometry, ICLR 2024.

[7] WEI-KAI CHANG and Rajiv Khanna, A Unified Stability Analysis of SAM vs SGD: Role of Data Coherence and Emergence of Simplicity Bias, NeurIPS, 2025

[8] Hao Li. Visualizing the loss landscape of neural nets. NeurIPS, 2018.

[9] Sidak Pal Singh. Woodfisher: Efficient second-order approximation for neural network compression. In NeurIPS, 2020

[10] Behrooz Ghorbani. An investigation into neural net optimization viahessian eigenvalue density. In International Conference on Machine Learning, PMLR,2019.

[11] Aditya Golatkar. Eternal sunshine of the spotless net: Selective forgetting in deep networks. CVPR, 2020.

[12] M. Ding, “Understanding fine-tuning in approximate unlearning: A theoretical perspective,” 2025.

[13] Pol Puigdemont, Ascent Fails to Forget. Ioannis Mavrothalassitis,  Nips 2025

[14] Anvith Thudi, Gabriel Deza, Varun Chandrasekaran, and Nicolas Papernot. Unrolling sgd: Understanding factors
influencing machine unlearning. IEEE, 2022.

---

> ### Author Response · Authors · 2025-11-22
>
> 2. Definition of the unlearning and divergence.
>
> We thank all reviewers for raising concerns about whether divergence or distance-based quantities are appropriate for analyzing unlearning. We address these concerns in two parts.
>
> First, divergence and distance remain standard, widely used unlearning metrics. There is currently no single universal definition of unlearning across domains: some applications emphasize privacy (MIA resistance), others focus on confusion removal, bias removal, safety, or utility preservation. Despite this diversity, loss and distance based metrics remain among the most commonly used evaluation tools, and our work is consistent with this long line of literature. Our use of the metric is consistent with many established works that continue to evaluate unlearning effectiveness using forget-set loss or accuracy. Examples include unlearning accuracy [1], forgetting error [2], forget performance [3], forgetting loss [4, 5] and distance-based measures [6]. They are also work characterizing the distance between solutions resulting from different algorithms [7]. These metrics are still used because they capture the effect of removing the forget set and they correlate strongly with practical privacy risks (e.g., MIA success). In this sense, analyzing divergence or distance quantities that govern how the model leaves the old solution is aligned with standard practice.
>
> Second, divergence is theoretically meaningful and aligns with prior unlearning theory. Several foundational works analyze unlearning by studying optimization trajectory and its reaction to different component involved in unlearning. For example, [8] study unlearning through analyzing drift of optimization trajectories built on quadratic loss and determine the weight difference at infinity time limit. [9] use linear models and weight-space distances to characterize approximate unlearning. Specifically, they use loss as criterion to quantify the difference brought by unlearning process. The recent work [7] operates in linear logistic regression, by studying closed solution based on the logistic loss, they describe the unlearning process through the weight difference between original one and the unlearned one. [10] analyze unlearning by unrolling the SGD recursion, which again relies on linearization of the gradient dynamics. Also their definition of unlearning error directly use the difference in weight space and connect the MIA attack to this theoretically defined quantity.
>
> [1] C. Fan, J. Liu, Y. Zhang, D. Wei, E. Wong, and S. Liu, “Salun: Empowering machine unlearning via gradient-based weight saliency in both image classification and generation,” ArXiv,2023.
>
> [2] Meghdad Kurmanji, Peter Triantafillou, and Eleni Triantafillou. Towards unbounded machine
> unlearning. NeurIPS, 2023.
>
> [3] Nathaniel Li, Alexander Pan, Anjali Gopal, Summer Yue, Daniel Berrios, Alice Gatti, Justin D Li, Ann-Kathrin Dombrowski, Shashwat Goel, Long Phan, et al. The WMDP Benchmark: Measuring and Reducing Malicious Use With Unlearning. arXiv preprint arXiv, 2024.
>
> [4] Laura Graves, Vineel Nagisetty, and Vijay Ganesh. Amnesiac machine learning. In Proceedings of the AAAI Conference on Artificial Intelligence, 2021.
>
> [5] Aditya Golatkar, Alessandro Achille, and Stefano Soatto. Eternal sunshine of the spotless net: Selective forgetting in deep networks. In Proceedings of the IEEE/CVF Conference on Computer Vision and Pattern Recognition, 2020.
>
> [6] Anvith Thudi, Gabriel Deza, Varun Chandrasekaran, and Nicolas Papernot. Unrolling sgd: Understanding factors influencing machine unlearning. In 2022 IEEE 7th European Symposium on Security and Privacy (EuroSP). IEEE, 2022.
>
> [7] Ascent Fails to Forget Ioannis Mavrothalassitis, Pol Puigdemont, Noam Itzhak Levi, Volkan
> Cevher, Nips 2025
>
> [8] Aditya Golatkar, Alessandro Achille, and Stefano Soatto. Eternal sunshine of the spotless net: Selective forgetting in deep networks. In Proceedings of the IEEE/CVF Conference on Computer Vision and Pattern Recognition, 2020.
>
> [9] M. Ding, R. Sharma, C. Chen, J. Xu, and K. Ji, “Understanding fine-tuning in approximate unlearning: A theoretical perspective,” 2025.
>
> [10] Anvith Thudi, Gabriel Deza, Varun Chandrasekaran, and Nicolas Papernot. Unrolling sgd: Understanding factors
> influencing machine unlearning. In 2022 IEEE 7th European Symposium on Security and Privacy (EuroSP), pp.
> 303–319. IEEE, 2022.

---

> > ### Author Response · Authors · 2025-11-22
> >
> > 3. Feasibility of computing data coherence
> >
> > A practical concern raised by reviewers is that the theory’s “central metric relies on per-sample Hessians or Gram matrices,” which are expensive or infeasible to compute on modern large models. This is a valid point. Exact per-sample Hessians in a deep network can be enormous but it is not a fatal flaw of the approach. There is strong precedent in ML theory where initially intractable quantities inspired new insights and eventually yielded practical approximations. The Fisher Information Matrix (FIM) and the full Hessian of a network are classic examples: early theoretical research treated them as important objects despite their size, and this spurred the development of methods to approximate or constrain them[1,2]. Natural gradient methods and second-order optimizers like K-FAC (Kronecker-Factored Approximate Curvature)[1] explicitly approximate the Fisher/Hessian to achieve near-optimal descent directions. In fact, Sharpness-Aware Minimization (SAM)[3] (a recent regularizer that improves generalization) was inspired by the idea of penalizing the Hessian’s largest eigenvalues (i.e. minimizing sharpness). SAM doesn’t compute the full Hessian; it uses a clever first-order approximation (perturbing weights to measure curvature indirectly), yet it stemmed from the principle that the Hessian spectrum matters. Likewise, the Neural Tangent Kernel was originally an N×N Gram matrix over data points which is seemingly impractical beyond toy datasets, but it led to kernel proxies and inspired practical diagnostics. For instance, researchers developed ways to estimate the NTK[4] or related Gram matrices for subsets of data to monitor training dynamics, and used the constant-NTK theory to justify why wide networks behave more predictably.
> >
> > In our case, Hessian alignment/coherence is introduced as a conceptual tool to understand unlearning. While we indeed computed it in a small controlled CNN to validate the theory, this is akin to how many theoretical analyses proceed: first verify on a “toy” setup where the exact metrics can be computed for clarity, then later work on scaling it up. It’s worth noting that many large-scale theoretical studies have found ways to approximate Hessian-based measures. For example, [5] developed numerical linear algebra techniques to estimate the entire Hessian eigenvalue density for ImageNet-scale networks. They used random matrix sketching and power-iteration methods to produce the Hessian spectrum efficiently, a feat that seemed impossible a few years prior. This underscores that what’s “infeasible” with brute force can become feasible with algorithmic ingenuity. The history of deep learning research shows that what starts as “only explanatory” can become actionable. The NTK was once purely theoretical, yet now practitioners talk about “NTK conditioning” and use kernel analogies to choose architectures. Likewise, we anticipate that coherence measures could inspire new diagnostics (perhaps a coherence score computed on a small held-out batch as a proxy) or new training procedures (e.g. encourage decorrelation between forget and retain gradients). In our submission, we acknowledged that directly computing per-sample Hessians for a giant model is impractical today, but we intentionally validated our theory in a setting where we could compute them exactly, thus establishing a clear ground truth. This is a valuable first step. Moving forward, our framework can guide research into scalable approximations: perhaps using low-rank factorization of the Hessian, or computing block-wise coherence (layer-wise, or between specific neural units) as a cheaper metric. Thus, we confidently defend the use of linear stability and local linearization in our analysis: it is a principled approach grounded in prior successes in deep learning theory, and it offers a powerful explanatory framework for understanding when models will – or won’t – let go of what they have learned.
> >
> > [1] James Martens and Roger Grosse. Optimizing neural networks with kronecker-factored
> > approximate curvature. In International conference on machine learning, pages 2408–2417.
> > PMLR, 2015.
> >
> > [2] Zhewei Yao, Amir Gholami, Kurt Keutzer, and Michael W Mahoney. Pyhessian: Neural
> > networks through the lens of the hessian. In 2020 IEEE international conference on big data
> > (Big data), pages 581–590, 2020.
> >
> > [3] Pierre Foret, Ariel Kleiner, Hossein Mobahi, and Behnam Neyshabur. Sharpness-aware
> > minimization for efficiently improving generalization. In International Conference on Learning Representations, 2021.
> >
> > [4] Roman Novak. Neural tangents: Fast and easy infinite neural networks in python. In International Conference on Learning Representations, 2020.
> >
> > [5] Behrooz Ghorbani. An investigation into neural net optimization via hessian eigenvalue density. In International Conference on Machine Learning, PMLR,2019.

---

### Author Response · Authors · 2025-11-25
**Revision update**

We thank all reviewers for their constructive feedback. We have uploaded a revised version of the manuscript that directly addresses the major concerns raised. Below we summarize the key clarifications and additions introduced in the revision.
We thank all reviewers for their constructive feedback. We have uploaded a revised version of the manuscript that directly addresses the major concerns raised. We emphasize that our paper is theoretical and well-grounded within the existing theoretical frameworks while extending them to unlearning settings. Below we summarize the key clarifications and additions introduced in the revision.

1. Clarifications on assumptions.
Several reviewers requested greater exposition regarding the linear (quadratic) approximation and the vanishing of samplewise gradients at interpolation. We clarify in following. First, formalization of the local quadratic model around minima. Second, we expand on the interpolation assumption and its empirical justification. Third, we demonstrate these assumptions within established linear-stability analyses of SGD. This clarifies the scope of the theory as a characterization of local stability near an interpolating minimum. (section 3.1 and section 5.3)

2. Explicit discussion of limitations.
We directly acknowledge the limitations of the approximation and clarify that our results describe the onset of instability, rather than the entire nonlinear unlearning trajectory. We also reference prior work using similar approximations to show how our assumptions align with established modeling practice, and is an important step towards more general analysis in the future, as is common in theoretical studies.

3. Expanded related work discussion including modern unlearning theory.
The revised related works section now discusses how our framework connect to prior works and discuss different framework attempted throughout the line of literature. This clarifies and motivate the goal of our work which is to study the optimization behavior through the data geometry in the loss landscape. (section 2 in second paragraph)

4. New CIFAR-10 + ResNet-18 experiments.
To address concerns regarding real-world applicability, we added experiments on CIFAR-10 with ResNet-18 using SNR-controlled memorization. Our results show that stronger memorization (low SNR) leads to faster divergence during unlearning, and weaker memorization (high SNR) leads to greater resistance, which are in agreement with the theoretical stability prediction.(section 4 in last paragraph)

5. Clarifying divergence vs. unlearning success.
We explain that the relevance divergence in our framework an indicator of escaping the original basin as a proxy for unlearning data that was learnt when the optimizer settled in the basin. This complements practical metrics such as membership inference attack, and establishes a foundational framework towards understanding theoretical limits of resistance to unlearning. Our theory captures the optimization theoretic constraints on when unlearning becomes possible through the lens of data geometry and curvature alignment. (section 5.4)

6. Practical feasibility of per-sample Hessians.
We discuss why the use of per-sample Hessians is conceptually justified, citing precedents such as natural gradients, K-FAC, Hessian-spectrum estimation methods, and NTK approximations. These examples show how theoretically intractable matrices have historically led to practical approximations, and how our framework fits on this trajectory. (section 5.5)

7. Expanded details of experimental design .
We provide further details into how the signal-to-noise ratio (SNR) is manipulated (by scaling the signal), the motivation for the synthetic setup, and the explanation toward underneath purposes for all quantities we tracked in our experiments. Additions are highlighted in blue in the revised manuscript. (section 5.3)

8. New discussion on types of memorization.
Because reviewers raised questions about LLM-style verbatim memorization, we added a section that distinguishes orthogonal memorization (our setting) from verbatim sequence memorization, cites recent evidence that memorized examples correspond to highly orthogonal directions, and identifies extending our geometric framework to sequential memorization in LLMs as an important open direction. (section 5.6)

Overall, the revised manuscript clarifies the assumptions, expands theoretical context, provides new experimental validation, and offers additional discussion to help readers understand the scope and implications of our contributions.

---

### Meta-Review · Area_Chair_GG5Q · 2025-12-12

**Summary:**

Across all four reviews, the main reason this paper appears below the bar is that despite an interesting theoretical framing, the scope, assumptions and empirical validation were initially seen as too limited to convincingly support the broader claims about “why models resist unlearning,” and the practical relevance and actionable takeaway was not sufficiently clear.

Concretely, several reviewers repeatedly questioned whether the analysis, which was built around local, linear or quadratic approximations and curvature and Hessian-style quantities, adequately represents realistic deep-net unlearning dynamics and whether the proposed metrics and definitions align with how “successful unlearning” is evaluated in practice such as privacy, forgetting vs. retention tradeoffs, etc.. In addition, there were presentation, positioning and related-work concerns: for example, Reviewer j7DZ flagged that the “findings” section and experimental design/variables (e.g., SNR) were confusing and could undermine interpretability. Reviewer MYQ6 raised that the introduction’s framing around “ascent-based unlearning” could be misleading relative to the broader unlearning literature and also noted technical/formatting issues and questions about a Taylor-expansion step.

Overall, even granting the paper’s theoretical interest, the consensus concerns centered on (a) realism of assumptions, (b) alignment with practical unlearning definitions/metrics, and (c) insufficient evidence that the proposed mechanism explains unlearning resistance broadly.

**Reviewer Concerns:**

Concerns addressed by the rebuttal
- The authors' “revision update” and responses explicitly clarify modeling assumptions on linear/quadratic approximation, stability conditions, the scope, and how they intend “divergence” or “distance”-style notions to be used.
- The authors report adding a CIFAR-10 + ResNet-18 experiment (training to convergence; probing memorization/unlearning correlation) and provide a table of results across settings. This likely improves confidence for at least one reviewer who explicitly asked for it.
- Reviewer j7DZ’s confusion around experiment purpose and variables (SNR) is directly acknowledged, with a claim that the design was clarified and some discussion structure updated.
- Reviewer MYQ6’s points about “ascent-based unlearning” framing and broken references were acknowledged. The authors state they will revise the intro and related-work framing and fix reference formatting.

Concerns still outstanding or only partially resolved
- The theory is still best-supported for local behavior near minima and for settings where curvature/Hessian-style proxies are meaningful. Even with clarifications and one added ResNet/CIFAR experiment, the reviewers’ broader worry on how far this mechanism extrapolates to modern unlearning pipelines and architectures, is only partially answered. Reviewer G3DN’s critique emphasized fragility of the approximation, assumptions and questioned whether the framework yields prescriptive and practically aligned guarantees.
- Reviewer 3vnF explicitly challenged the definition and metric choice, and how “success” relates to forgetting vs. retention in a practically meaningful way. The rebuttal clarifies intent, but it doesn’t fully resolve the concern that the paper may be optimizing or explaining a notion that doesn’t map cleanly to common unlearning objectives across domains.
- A recurring undercurrent is that the work reads as an explanatory lens (“why stuck near minima”) more than a method or diagnostic that practitioners can reliably use. The added experiment helps, but the “what should I do differently when unlearning fails?” takeaway is still not sharply established, particularly across diverse unlearning methods/settings.

**Reviewer Scores:**

Given all four reviews were 4, the rebuttal seems to have improved clarity and added one stronger experiment, but it likely would not have been enough to overcome the remaining scope and practical-alignment concerns for most reviewers.

Reviewer 3vnF (score: 4 $\rightarrow$ likely 4): Reviewer 3vnF’s main issues were conceptual (appropriateness of curvature alignment mechanism, coherence, and metric choice). The rebuttal responds, but the foundational skepticism likely remains, so I’d expect no score change.

Reviewer G3DN (score: 4 $\rightarrow$ likely 4 or 6; I’d predict 4): This reviewer asked explicitly for more realistic experiments (ResNet/CIFAR-10).  The authors added such an experiment.  This could justify a move to 6 if the reviewer felt the new results strongly supported the thesis; however, because the review also raised multiple broader limitations (assumptions, “successful unlearning” definition/practicality, computational feasibility), my best guess is stays at 4 (maybe a “positive 4”).

Reviewer j7DZ (score: 4 $\rightarrow$ likely 6): j7DZ’s criticisms look more “fixable” (organization, clarity of experiments/variables, why SNR is fixed, etc.).  Since the rebuttal directly acknowledges confusion and claims concrete clarifications/updates, this reviewer is the most likely to be persuaded upward to a 6.

Reviewer MYQ6 (score: 4 $\rightarrow$ likely 4): MYQ6 had a mix of related-work framing concerns and technical/writeup issues (ascent-based framing, typos/references, Taylor-expansion detail). The authors agree to revise the framing and fix references and answer the technical point. Still, because the reviewer also questioned whether the analysis generalizes beyond the specific setting, I’d expect no score change.

---

### Decision · Program_Chairs · 2026-01-26

Reject